# Towards identifiability of micro total effects in summary causal graphs with latent confounding: extension of the front-door criterion

**Charles K. Assaad**                                                    *charles.assaad@inserm.fr*
*Sorbonne Université, INSERM*
*Institut Pierre Louis d'Epidémiologie et de Santé Publique*
*F75012, Paris, France*

**Reviewed on OpenReview:** *https://openreview.net/forum?id=5f7YlSKG1l*

## Abstract

Conducting experiments to estimate total effects can be challenging due to cost, ethical concerns, or practical limitations. As an alternative, researchers often rely on causal graphs to determine whether these effects can be identified from observational data. Identifying total effects in fully specified causal graphs has received considerable attention, with Pearl's front-door criterion enabling the identification of total effects in the presence of latent confounding even when no variable set is sufficient for adjustment. However, specifying a complete causal graph is challenging in many domains. Extending these identifiability results to partially specified graphs is crucial, particularly in dynamic systems where causal relationships evolve over time. This paper addresses the challenge of identifying total effects using a specific and well-known partially specified graph in dynamic systems called a summary causal graph, which does not specify the temporal lag between causal relations and can contain cycles. In particular, this paper presents sufficient graphical conditions for identifying total effects from observational data, even in the presence of cycles and latent confounding, and when no variable set is sufficient for adjustment.

## 1  Introduction

Causal questions arise when we seek to understand the effects of interventions, such as asking, "If we administer today a hypertension treatment to a patient with kidney insufficiency, will the kidney function (represented by the creatinine level) improve tomorrow?". These questions, often referred to as total causal effects or simply total effects (Pearl et al., 2000), are denoted[1] as $\Pr(Creatinine_{tomorrow} = c \mid do(Hypertension_{today} = h))$ where the do() operator denotes an intervention. They differ from associational relationships, $\Pr(Creatinine_{tomorrow} = c \mid Hypertension_{today} = h)$, as they isolate the effect of an intervention, disregarding other influencing factors such as sodium intake or protein intake or stress level. Experimentation is known as the traditional approach across various fields to estimate the total effect of interventions free from confounding bias (Neyman et al., 1990). However, conducting experiments is not always feasible due to cost, ethical considerations, or practical limitations. Consequently, scientists often resort to estimating effects of interventions from observational data. This process relies on specific assumptions and typically involves two sequential steps: identifiability and estimation (Pearl, 2019). The identifiability step refers to the question of whether the total effect of interest can be uniquely determined from the available data and the assumptions made about the causal model. This step usually also include finding a way to express the total effect in terms of the observed data distribution, i.e., using a do-free expression. On the

---

[1]In a nonparametric setting, the total effects is a specific functional of $\Pr(Creatinine_{tomorrow} = c \mid do(Hypertension_{today} = h))$ for different values of $h$. However, for simplicity, the total effect is often reffered to as $\Pr(Creatinine_{tomorrow} = c \mid do(Hypertension_{today} = h))$.

other hand, the estimation step refers to the process of calculating the value of a total effect, once it has been identified, from finite observational data using statistical methods. This paper focuses on the first step.

Graphical models, provide a framework for identifying total effects from causal graphs which encode variables as vertices and causal relationships as arrows, allowing researchers to visualize and analyze complex causal structures (Pearl et al., 2000). Identifying total effects in fully specified non-temporal causal graphs has been a subject of considerable attention (Pearl, 1993b; 1995; Spirtes et al., 2000; Pearl et al., 2000; Shpitser & Pearl, 2008; Shpitser et al., 2010). Adjusting for covariates is one method among several that allow us to identify causal effects and the back-door criterion (Pearl, 1993b) is one of the most known methods that allow us to find covariates using causal graphs. Pearl (1993a; 1995) has provided examples where no set of variables is sufficient for adjustment, yet the causal effect can still be consistently estimated through multi-stage adjustments. Pearl's front-door criterion offers a graphical method for identifying total effects despite the absence of an adjustment set due to latent confounding, assuming the causal graph is a directed acyclic graph (DAG) or an acyclic directed mixed graph (ADMG). Initially criticized for its limited practical application, this criterion has recently gained recognition and is now employed in epidemiology (Inoue et al., 2022; Piccininni et al., 2023).

The above identifiability methods are directly applicable to fully specified temporal graphs (Blondel et al., 2016) which represent causal relations in dynamic systems where causal relationships evolve over time. However, constructing a fully scpecified temporal graph requires knowledge of all causal relationships among observed variables, which is often unavailable, especially in many real-world applications. However, experts may know that one variable causes another without knowing the exact temporal lag. For example, understanding the transmission of SARS-CoV-2 from younger to older individuals, and vice versa, can help define interventions most likely to reduce the number of deaths. Indeed, it has been shown that younger adults tended to be highly infected during the first wave of the pandemic, while older individuals faced a higher risk of death if infected (Carrat et al., 2021; Lapidus et al., 2021; Glemain et al., 2024). Considering sufficiently large time intervals (several weeks) as in repeated serosurveys, like in Wiegand et al. (2023), it is not clear if the number of new infections in one age group during a time interval (incidence) may be influenced by incidence in the other age group during the same interval. Incidence in an age group can also be influenced by incidence during the previous time interval in any age group. Therefore, constructing a fully specified causal graph is difficult. In such cases, partially specified causal graphs can be useful. A very well known and useful partially specified causal graph is the summary causal graph (SCG) which represents causal relations without including temporal information, i.e., each vertex represents a time series. Both medical and epidemiological examples given above can be represented by an SCG with a cyclic relationship representing the interplay between creatinine and hypertension in the first example and between the two age groups in the second example.

Recently, there has been new interest in extending identifiability results to partially specified graphs Eichler & Didelez (2007; 2010); Maathuis & Colombo (2013); Perkovic (2020); Wang et al. (2023); Anand et al. (2023); Ferreira & Assaad (2024); Assaad et al. (2024); Ferreira & Assaad (2025). Most of these partially specified graphs represent Markov equivalence classes, where the partial specifications differ conceptually from those in SCGs. For instance, in these graphs, partial specification typically manifests as undirected edges or edges with specific endpoints indicating uncertainty about the orientation. Additionally, each vertex in these graphs corresponds directly to an observed variable, maintaining a straightforward one-to-one relationship. As a result, the extensions of identifiability results for these graphs (Maathuis & Colombo, 2013; Perkovic, 2020; Wang et al., 2023) are not applicable to SCGs. Another important type of partially specified graph is the cluster graph, which represents causal relationships between clusters of variables rather than individual variables. This means that, unlike graphs representing Markov equivalence classes, the skeleton of a cluster graph does not correspond to the skeleton of true causal graph. SCGs are a specific type of cluster graphs, where each cluster represents a time series. Most works extending identifiability results to cluster graphs have focused on extreme multivariate cases, where the goal is to identify the total effect of one entire cluster on another entire cluster (Anand et al., 2023; Ferreira & Assaad, 2025). The few studies that consider the total effect of a single variable within a cluster on another single variable within a different cluster have been conducted on SCGs. For example, under the assumptions of no instantaneous relations and no latent confounding between two timepoints within the same time series, Eichler & Didelez (2007; 2010)

demonstrated that both the back-door and front-door criteria are applicable in SCGs. Assaad et al. (2023) further established that in the absence of latent confounding, the total effect remains identifiable through adjustment in acyclic SCGs, even when instantaneous relations are present. More recently, Assaad et al. (2024) provided graphical conditions for identifying total effects via adjustment in SCGs that incorporate both cycles and instantaneous relations, still under the assumption of no latent confounding. Finally, independently and concurrently of this work , Reiter et al. (2024) investigated the identification of total effects in the frequency domain using the SCG (referred to as a process graph in their work) under the assumption of linearity; however, unlike this and previous works (Eichler & Didelez, 2007; 2010; Assaad et al., 2023; 2024), they did not explore specific graphical conditions within the SCG for identification.

The previous paragraph underscores the novelty of this work: in the setting where a fully specified temporal causal graph is not available, this paper is the first to address the challenge of nonparametrically and graphically identifying total effects of *one variable within a cluster* on *another variable in a different cluster* when having access to an SCG while allowing *instantaneous relations*, *cycles*, and *latent confounding* (even between two timepoints within the same time series). It shows that the standard front-door criterion (Pearl, 1993a; 1995) and its previous extension to time-series (Eichler & Didelez, 2007; 2010) are not sound when applied to SCGs in the general setting considered in this paper. Nevertheless, by leveraging the front-door criterion, this work introduces sufficient conditions to identify total effects from observational data using SCGs in scenarios where latent confounding prevents identifiability through standard adjustment methods.

The remainder of the paper is organized as follows: Section 2 introduces necessary terminology and tools and it formalizes the problem. Section 3 demonstrates that the standard front-door criterion and its previous extension to time series are unsuitable when applied to SCGs. Section 4 presents the main result of this paper and Section 5 discusses several examples of non-identifiability. Finally, Section 6 concludes the paper.

## 2 Preliminaries and problem setup

This section, first introduces some terminology and tools which are standard for the major part and then, formalize the problem that this paper is going to solve. Let us start by defining the causal model that is considered.

**Definition 1** (Discrete-time dynamic structural causal model (DTDSCM)). *A discrete-time dynamic structural causal model is a tuple $\mathcal{M} = (\mathbb{L}, \mathbb{V}, \mathbb{F}, P(\mathbb{I}))$, where $\mathbb{L} = \bigcup \{\mathbb{L}^{v_t^i} \mid i \in [1, d], t \in [t_0, t_{max}]\}$ is a set of exogenous variables, which cannot be observed but affect the rest of the model. $\mathbb{V} = \bigcup \{\mathbb{V}^i \mid i \in [1, d]\}$ such that $\forall i \in [1, d]$, $\mathbb{V}^i = \{V_t^i \mid t \in [t_0, t_{max}]\}$, is a set of endogenous variables, which are observed and every $V_t^i \in \mathbb{V}$ is functionally dependent on (directly caused by) some subset of $\mathbb{L}^{v_t^i} \cup \mathbb{V}_{\leq t} \setminus \{V_t^i\}$ where $\mathbb{V}_{\leq t} = \{V_{t'}^j \mid j \in [1, d], t' \leq t\}$. $\mathbb{F}$ is a set of functions such that for all $V_t^i \in \mathbb{V}$, $f^{v_t^i}$ is a mapping from $\mathbb{L}^{v_t^i}$ and a subset of $\mathbb{V}_{\leq t} \setminus \{V_t^i\}$ to $V_t^i$. $P(\mathbb{I})$ is a joint probability distribution over $\mathbb{L}$.*

A DTDSCM implicitly assumes that an effect cannot precede its cause. This assumption is explicitly stated as follows.

**Assumption 1.** *Consider a DTDSCM $\mathcal{M} = (\mathbb{L}, \mathbb{V}, \mathbb{F}, P(\mathbb{I}))$. Suppose $V_t^i$ is an endogenous variable which is functionally dependent on $\mathbb{W} \subseteq \mathbb{V} \setminus \{V_t^i\}$, i.e., $V_t^i := f^{v_t^i}(\mathbb{W}, \mathbb{L}^{v_t^i})$. For all $V_{t'}^j \in \mathbb{W} \cup \mathbb{L}^{v_t^i}$, it is assumed that $t' \leq t$.*

Furthermore, stationarity is assumed.

**Assumption 2.** *Consider a DTDSCM $\mathcal{M} = (\mathbb{L}, \mathbb{V}, \mathbb{F}, P(\mathbb{I}))$. $\forall f^{v_t^i}, f^{v_{t'}^i} \in \mathbb{F}$, $f^{v_t^i} = f^{v_{t'}^i}$.*

Assumption 2 have two important implications. First, it entails that if $Y_t = f^{y_t}(X_{t-1}, W_{t-1})$, then $\forall t' \in [t_0 + 1, t_{max}]$, $Y_{t'} = f^{y_t}(X_{t'-1}, W_{t'-1})$. Furthermore, it allows us to fix the maximum temporal lag between a cause and an effect, denoted as $\gamma_{max}$. The contributions of this paper are relevant whenever multivariate time series—where each $\mathbb{V}^i$ such that $i \in [1, d]$ represents a time series with a single observation per temporal variable $\mathbb{V}_t^i$—or multivariate spatio-temporal series —where each $\mathbb{V}^i$ such that $i \in [1, d]$ is a spatio-temporal series with multiple observations exist for each temporal variable $\mathbb{V}_t^i$ (as seen in certain types of cohort studies)—are considered. In the former case, if Assumption 2 is violated, identifying a unique total

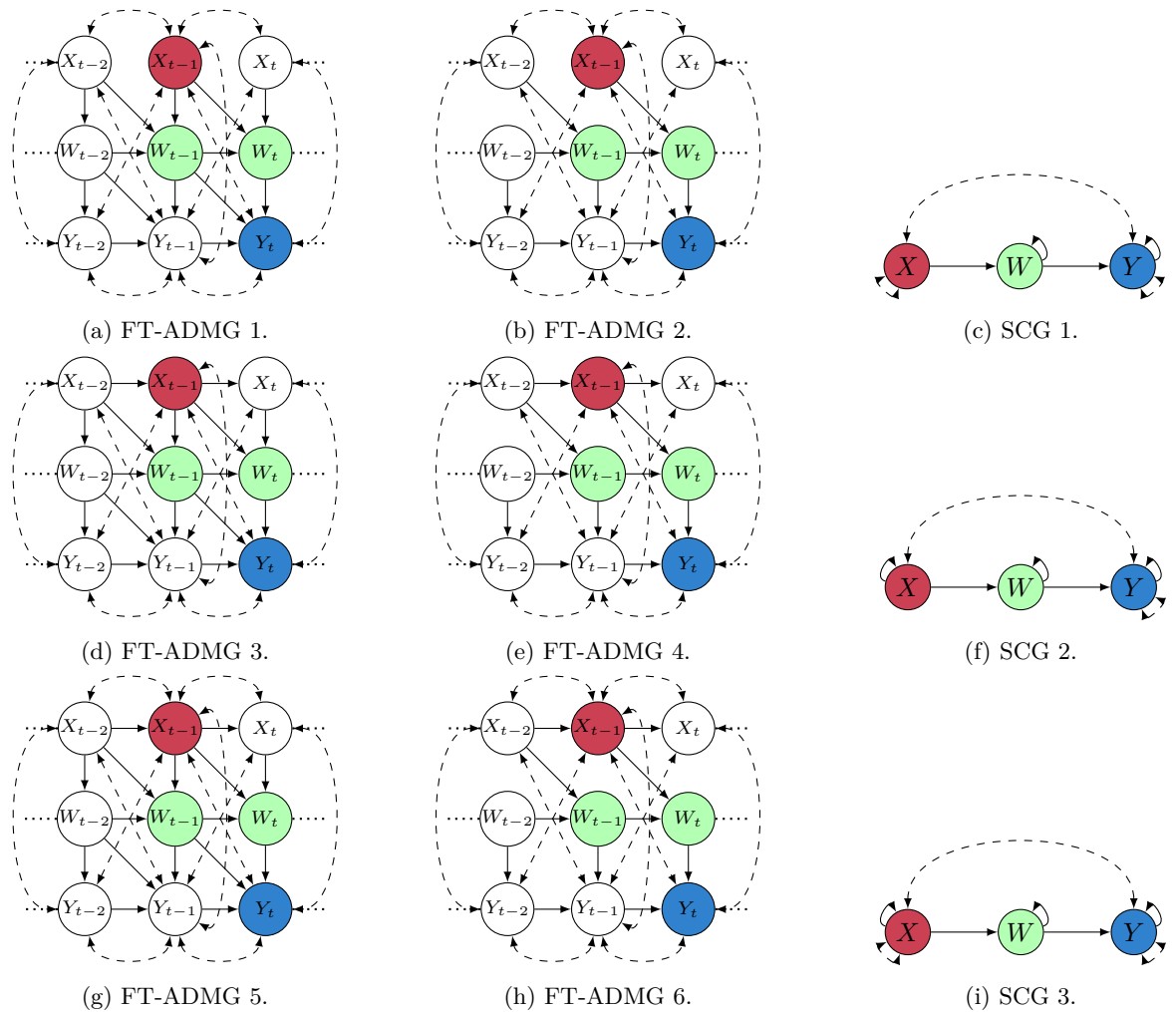

Figure 1: Three SCGs and six FT-ADMGs, where $\gamma_{\max} = 1$, such that FT-ADMGs 1 and 2 are compatible with SCG 1, FT-ADMGs 3 and 4 are compatible with SCG 2, and FT-ADMGs 5 and 6 are compatible with SCG 3. Each pair of red and blue vertices represents the cause and the effect of interest and green vertices are those that interpect all paths from the cause to the effect of interest. In SCG 1 and SCG 2, $\{W\}$ satisfies Definition 7 for the total effect $\Pr(y_t|do(x_{t-1}))$. However, $\{W\}$ does not satisfy Definition 7 in SCG 3 for the total effect $\Pr(y_t|do(x_{t-1}))$ since $Cycles(X, \mathcal{G}^s) \neq \emptyset$ and $\gamma \neq 0$.

effect becomes ill-posed. This is because this case assumes a dynamic system with a single multivariate observational time series, meaning that variables preceding a given variable $V_t^i$ within the same time series $\mathbb{V}^i$ would act as substitutes for different realizations of that variable. Violating this assumption would imply that the total effect varies over time, making estimation impossible. Furthermore, in this case, additional assumptions—such as those proposed in Eichler & Didelez (2010)—are required for the estimation of the total effect. However, in the latter case, where multiple observations of each temporal variable $\mathbb{V}_t^i$ are available, these additional assumptions are not necessary. The distinction between these two cases is considered to be beyond the scope of this work. Therefore, I will not explicitly differentiate between them and for clarity and brevity, I will refer to the set of variables $\mathbb{V}^i$ simply as a time series throughout the remainder of this paper.

It is also supposed that the DTDSCM can be qualitativly represented by full-time[2] directed acyclic graph (FT-DAG), commonly known as a full-time causal graph (Peters et al., 2013). To check the identifiability of

---

[2]The term "full-time" underscores that the graph represents the entirety of a dynamic system over time (from $t_0$ to $t_{max}$). This representation can stretch over a vast range, suggesting that any segment of a full-time DAG depicted in a figure should

a total effect, it is standard to first transform the FT-DAG to a full-time acyclic directed mixed graph (FT-ADMG) (Richardson, 2003) which is naturally derived from FT-DAGs with latent variables via an operation called latent projection (Tian & Pearl, 2002). The FT-ADMG is supposed to be a DAG with bidirected edges representing latent confounding. If all variables representing latent confounding become observed then the FT-ADMG becomes a FT-DAG. Figures 1a, 1b, 1d,1e, 1g and 1h present six different FT-ADMGs. In our setting, FT-ADMGs can also be obtained directly from the DTDSCM, as shown below.

**Definition 2** (Full-Time Acyclic Directed Mixed Graph). *Consider a DTDSCM $\mathcal{M}$. The* full-time acyclic directed mixed graph *(FT-ADMG) $\mathcal{G} = (\mathbb{V}, \mathbb{E})$ induced by $\mathcal{M}$ is defined in the following way:*

$$\mathbb{E}^1 := \{ X_{t'} \to Y_t \quad | \forall Y_t \in \mathbb{V},\ X_{t'} \in \mathbb{X} \text{ such that } Y_t := f_t^y(\mathbb{X}, \mathbb{L}^{y_t}) \text{ in } \mathcal{M} \text{ and } \mathbb{X} \subset \mathbb{V} \backslash \{Y_t\} \},$$

$$\mathbb{E}^2 := \{ X_{t'} \longleftrightarrow Y_t \quad | \forall X_{t'}, Y_t \in \mathbb{V} \text{ such that } \mathbb{L}^{x_{t'}} \not\!\perp\!\!\!\perp \mathbb{L}^{y_t} \}.$$

*where $\mathbb{E} = \mathbb{E}^1 \cup \mathbb{E}^2$.*

**FT-ADMG notions** For an FT-ADMG $\mathcal{G}$, a *path* from $X_{t'}$ to $Y_t$ in $\mathcal{G}$ is a sequence of distinct vertices $\langle X_{t'}, \dots, Y_t \rangle$ in which every pair of successive vertices is adjacent. A *directed path* from $X_{t'}$ to $Y_t$ is a path from $X_{t'}$ to $Y_t$ in which all edges are directed towards $Y_t$ in $\mathcal{G}$, that is $X_{t'} \to \dots \to Y_t$. A *back-door path* between $X_{t'}$ and $Y_t$ is a path between $X_{t'}$ and $Y_t$ with an arrowhead into $X_{t'}$ in $\mathcal{G}$. If $X_{t'} \to Y$, then $X_{t'}$ is a *parent* of $Y_t$. If there is a directed path from $X_{t'}$ to $Y_t$, then $X_{t'}$ is an *ancestor* of $Y_t$, and $Y_t$ is a *descendant* of $X_{t'}$. A vertex counts as its own descendant and as its own ancestor. The sets of parents, ancestors and descendants of $X_{t'}$ in $\mathcal{G}$ are denoted by $\text{Par}(X_{t'}, \mathcal{G})$, $\text{Anc}(X_{t'}, \mathcal{G})$ and $\text{Desc}(X_{t'}, \mathcal{G})$ respectively. If a path $\pi$ contains $X_{t'} \to W_{t''} \leftarrow Y_t$ as a subpath, then $W_{t''}$ is a *collider* on $\pi$. A path $\pi$ from $X_{t'}$ to $Y_t$ is *active* given a vertex set $\mathbb{W}$, with $X_{t'}, Y_t \notin \mathbb{W}$ if every non-collider on $\pi$ is not in $\mathbb{W}$, and every collider on $\pi$ has a descendant in $\mathbb{W}$. Otherwise, $\mathbb{W}$ *blocks* $\pi$. A set of vertices $\mathbb{W}$ intercepts all directed paths from $X_{t'}$ to $Y_t$ if every directed path from $X_{t'}$ to $Y_t$ passes through at least one vertex in $\mathbb{W}$. Lastly, each vertex in an FT-ADMG is called a temporal vertex or a micro vertex.

The *total effect* (Pearl et al., 2000) between two micro variables is written as $P(Y_t = y_t | do(X_{t-\gamma} = x_{t-\gamma}))$. $Y_t$ corresponds to the response and $do(X_{t-\gamma} = x_{t-\gamma})$ represents an intervention (as defined in Pearl et al. (2000) and Eichler & Didelez (2007, Assumption 2.3)) on the variable $X$ at time $t - \gamma$, with $\gamma \geq 0$. Unlike $\gamma_{max}$ (which represents the maximum possible lag between a cause and effect), $\gamma$ simply specifies the lag of interest for the query posed by the user. In the remainder of the paper, $\gamma$ is considered to be in $\{0, \gamma_{\max}\}$, and, with a slight abuse of notation, $P(Y_t = y_t \mid do(X_{t-\gamma} = x_{t-\gamma}))$ is written as $P(y_t \mid do(x_{t-\gamma}))$. The identifiability of the total effect in FT-ADMGs is defined as follows.

**Definition 3** (Identifiability of total effects in FT-ADMGs). *Let $X_{t-\gamma}$ and $Y_t$ be distinct vertices in an FT-ADMG $\mathcal{G} = (\mathbb{V}, \mathbb{E})$. The total effect of $X_{t-\gamma}$ on $Y_t$ is identifiable in $\mathcal{G}$ if $\Pr(y_t | do(x_{t-\gamma}))$ is uniquely computable from any positive observational distribution consistent with $\mathcal{G}$.*

A total effect is uniquely computable if $\Pr(y_t \mid do(x_{t-\gamma}))$ can be expressed using a *do-free formula*. Given a fully specified FT-ADMG, there exists many tools to identify the total effect. For example the standard backoor criterion (Pearl, 1993b; 1995) can be used to find a set of covariates $\mathbb{B}$ that is sufficient for adjustment; in such case the do-free formula of the total effect is written as $\sum_{\mathbb{b}} \Pr(y_t \mid x_{t-\gamma}, \mathbb{b}) \Pr(\mathbb{b})$. When such a set does not exist due to latent confounding, the standard front-door criterion, as introduced by Pearl (1995), can sometimes enable the derivation of an alternative do-free formula.

However, in many real-world applications such as medicine or epidemiology, experts often cannot provide the FT-ADMG. Furthermore, discovering the true FT-ADMG from real observational data is challenging (Spirtes et al., 2000; Mogensen et al., 2018; Runge et al., 2019; Assaad et al., 2022) because 1) causal discovery algorithms rely on untestable assumptions that are not always satisfied in real applications (Aït-Bachir et al., 2023) and 2) even when the assumptions hold, the output of such methods does not always correspond to the true graph but rather to a class of graphs that include the true graph (Gerhardus & Runge, 2020). Therefore,

---

be viewed as merely a snapshot of the larger graph. Under Assumption 2, it is sometimes possible to extrapolate the entire graph from such a snapshot.

experts typically rely on a partially specified representation of the FT-ADMG, known as a summary causal graph[3].

**Definition 4** (Summary Causal Graph with possible latent confounding). *Consider an FT-ADMG $\mathcal{G} = (\mathbb{V}, \mathbb{E})$. The summary causal graph (SCG) $\mathcal{G}^s = (\mathbb{S}, \mathbb{E}^s)$ compatible with $\mathcal{G}$ is defined in the following way:*

$$
\begin{aligned}
\mathbb{S} :=& \{V^i = (V^i_{t_0}, \cdots, V^i_{t_{max}}) & | \forall i \in [1, d]\}, \\
\mathbb{E}^{s1} :=& \{X \rightarrow Y & | \forall X, Y \in \mathbb{S}, \ \exists t' \leq t \in [t_0, t_{max}] \ such \ that \ X_{t'} \rightarrow Y_t \in \mathbb{E}\}, \\
\mathbb{E}^{s2} :=& \{X \leftarrow\!\cdots\!\rightarrow Y & | \forall X, Y \in \mathbb{S}, \ \exists t', t \in [t_0, t_{max}] \ such \ that \ X_{t'} \leftarrow\!\cdots\!\rightarrow Y_t \in \mathbb{E}\}.
\end{aligned}
$$

*where $\mathbb{E}^s = \mathbb{E}^{s1} \cup \mathbb{E}^{s2}$.*

**SCG notations** For an SCG $\mathcal{G}^s$, a directed path from $X$ to $Y$ and the edge $Y \rightarrow X$ form a *directed cycle* in $\mathcal{G}^s$. Additionally, the specific case where a vertex has a directed edge to itself (self loop) is considered as a cycle too. $Cycles(X, \mathcal{G}^s)$ denotes the set of all directed cycles containing $X$ in $\mathcal{G}^s$. A *directed path* between $X$ and $Y$ is a path between $X$ and $Y$ which starts by $X \rightarrow$ and does not contain any arrow on the path pointing strictly towards $X$. A *back-door path* between $X$ and $Y$ is a path between $X$ and $Y$ which starts by either $X \leftarrow$ or $X \leftarrow\!\cdots\!\rightarrow$ or $X \leftrightarrows$. If $X \rightarrow Y$ or $X \leftrightarrows Y$, then $X$ is a *parent* of $Y$. The notions of ancestors and descendants are defined similarly as in the case of FT-ADMGs. A path in an SCG is blocked given a set $\mathbb{W}$ if it contains a strict collider at $W$ (i.e., $\rightarrow W \leftarrow$, and not $\leftrightarrows W \leftarrow$ or $\rightarrow W \leftrightarrows$) such that $\mathbb{W} \cap Desc(W, \mathcal{G}^s) = \emptyset$ or if it contains strict non-collider at $W$ (i.e., $\rightarrow W \rightarrow$ or $\leftarrow W \rightarrow$ or $\leftarrow W \leftrightarrows$, and not $\leftrightarrows W \leftarrow$) such that $W \in \mathbb{W}$ and there exists a directed edge pointing from $W$ to a vertex on the path that does not form a cycle with $W$. A path in an SCG is activated if it is not blocked. In particular, a path is activated by an empty set if it does not contain any strict collider. The notions of interception is defined similarly as in the case of FT-ADMGs. Lastly, each vertex in an SCG is called a cluster or a macro vertex and it represents a time series.

Many FT-ADMGs might share the same compatible SCG. For example, Figure 1c presents the SCG compatible with the two FT-ADMGs in Figures 1a and 1b, Figure 1f presents the SCG compatible with the two FT-ADMGs in Figures 1d and 1e, and Figure 1i presents the SCG compatible with the two FT-ADMGs in Figures 1g and 1h. For a given SCG $\mathcal{G}^s$, any FT-ADMG from which $\mathcal{G}^s$ can be derived is called as a *candidate FT-ADMG* for $\mathcal{G}^s$. The set of all candidate FT-ADMGs for $\mathcal{G}^s$ is denoted by $\mathcal{C}(\mathcal{G}^s)$.

This paper focuses on identifying the total effect *when the only knowledge one has of the underlying DTDSCM consists in the SCG derived from the unknown, true FT-ADMG*. In this setting, the identifiability of the total effect in SCGs is defined as follows:

**Definition 5** (Identifiability of total effects in SCGs). *Consider an SCG $\mathcal{G}^s$. Let $X_{t-\gamma}$ and $Y_t$ be distinct vertices in every candidate FT-ADMG in $\mathcal{C}(\mathcal{G}^s)$. The total effect of $X_{t-\gamma}$ on $Y_t$ is identifiable in $\mathcal{G}^s$ if $\Pr(y_t | do(x_{t-\gamma}))$ is uniquely computable from any positive observational distribution consistent with any FT-ADMG in $\mathcal{C}(\mathcal{G}^s)$.*

Obviously, when the true FT-ADMG is unknown but the compatible SCG is accessible, it is possible to enumerate all candidates FT-ADMGs and then search for a do-free formula applicable for each of those FT-ADMGs. Within this approach, it is possible to identify the total effect if it is possible to find a set of micro vertices that satisfies the front-door criterion for each FT-ADMG in $\mathcal{C}(\mathcal{G}^s)$. However, enumerating all candidate FT-ADMGs is computationally expensive (Robinson, 1977), even when considering the constraints given by an SCG. Therefore, this paper addresses the following technical problem:

---

[3]One key motivation for employing summary causal graphs stems from the current limitations of causal discovery methods, which often struggle in practical applications due to their reliance on non-testable strong assumptions. Particularly in fields like medicine and epidemiology, researchers tend to prefer graphs built from prior knowledge rather than those inferred purely from data. However, fully specified graphs are very complicated to construct and validate manually and that is why it is important to work with (and ask experts to build) partially specified graphs such as summary causal graphs. That being said, recent studies have demonstrated that inferring summary causal graphs from data is more feasible than inferring FT-ADMGs from data (Wahl et al., 2024). This supports the relevance of our work even if researchers choose to utilize causal discovery, suggesting that our approach remains applicable when taking a data-driven approach to get the summary causal graph.

**Problem 1.** *Consider an SCG $\mathcal{G}^s$ and the total effect $\Pr(y_t|do(x_{t-\gamma}))$. The aim is to find sufficient conditions for identifying $\Pr(y_t|do(x_{t-\gamma}))$ using an SCG with latent confounding without enumerating all candidate FT-ADMGs in $\mathcal{C}(\mathcal{G}^s)$.*

## 3 Unsuitability of the standard front-door criterion when applied to SCGs

This section elucidates why the standard front-door criterion does not straightforwardly apply to SCGs. The standard front-door criterion consists of three conditions. Initially, the criterion was introduced for ADMGs (which means it can also be correctly applied to an FT-ADMG), but in order to illustrate its unsuitability in the context of this paper, it is presented here with few modification (in blue) given Definition 6.

**Definition 6** (Standard front-door criterion naively applied to SCGs)**.** *Consider an SCG $\mathcal{G}^s$. A set of macro vertices $\mathbb{W}$ in $\mathcal{G}^s$ satisfy the front-door criterion relative to a pair of micro vertices $(X_{t-\gamma}, Y_t)$ compatible with a pair of macro vertices $(X, Y)$ in $\mathcal{G}^s$ if:*

1. *$\mathbb{W}$ intercepts all activated directed paths from $X$ to $Y$;*
2. *there is no activated back-door path from $X$ to $\mathbb{W}$;*
3. *all back-door paths from $\mathbb{W}$ to $Y$ are blocked by $X$;*

To obtain the standard front-door criterion for FT-ADMGs, the terms in blue need to be modified as follows: since the variables of interest $(X_{t-\gamma}, Y_t)$ are already vertices in the given graph (FT-ADMG), there's no need to map these vertices to their compatible counterparts in the SCG. Therefore, the phrase "compatible with a pair of macro vertices $(X, Y)$ in $\mathcal{G}^s$" should be removed. Next, replace all remaining instances of "SCGs" with "FT-ADMGs" and "$\mathcal{G}^s$" with "$\mathcal{G}$". Additionally, since FT-ADMGs do not include macro vertices, the term "macro" should be replaced with "micro". Finally, replace all remaining instances of "$X$" and "$Y$" with "$X_{t-\gamma}$" and "$Y_t$". Pearl's insight is that if there exists a set $\mathbb{W}$ that intercepts all directed paths from $X_{t-\gamma}$ to $Y_t$ and there is no hidden confounding that cannot be blocked by $X_{t-\gamma}$, the total effect of $X_{t-\gamma}$ on $Y_t$ can be identified. This is achieved by: (i) identifying the effect of $X_{t-\gamma}$ on $\mathbb{W}$ (which is identifiable because the unobserved confounders influence $X_{t-\gamma}$ but not $\mathbb{W}$); (ii) identifying the effect of $\mathbb{W}$ on $Y_t$ conditional on $X_{t-\gamma}$ (which is identifiable because the unobserved confounders affect $Y_t$ but not $\mathbb{W}$); and (iii) multiplying the do-free formulas $\Pr(x_{t-\gamma} \mid \mathbb{w})$ and $\Pr(y_t \mid x_{t-\gamma}, \mathbb{w}) \Pr(x_{t-\gamma})$. Intuitively, in the context of an FT-ADMG (or an ADMG), the standard front-door criterion enables the identification of a total effect that cannot be identified through adjustment alone, by decomposing it into two identifiable total effects. The previously proposed extension of the front-door criterion to time series settings Eichler & Didelez (2010)–referred to here as the Granger front-door criterion–equivalent to the one given in Definition 6 with one minor change which makes slightly more restrictive: the Granger front-door criterion forbid all back-door paths from $X$ to $\mathbb{W}$ (activated and not activated) in Condition 2. In contrast to the criterion proposed in the next section, the Granger front-door criterion relies on two key assumptions: it implicitly assumes there is no latent confounding across different time points within a single time series, and it explicitly rules out instantaneous causal relationships. Under these assumptions–i.e., when the SCG reflects the absence of latent confounding over time and no instantaneous effects–the Granger front-door criterion allows identification of a total effect that is not identifiable via standard adjustment, by decomposing it into two identifiable total effects.

It turns out that directly applying the standard front-door criterion as introduced in Pearl (1995) to SCGs is not suitable. The main reason for this unsuitability is that not having a back-door path between two macro vertices in the SCG does not imply that there is no back-door path between two compatible micro vertice in any FT-ADMG in $\mathcal{C}(\mathcal{G}^s)$. Which means that even if Conditions 2 and 3 of Definition 6 are satisfied in the SCG $\mathcal{G}^s$, this does not guarantee that Conditions 2 and 3 of the standard front-door criterion are met for any FT-ADMG within $\mathcal{C}(\mathcal{G}^s)$. Sometimes, this would imply that even if the standard front-door criterion is satisfied when applied to SCGs, the total effect of interest might be non identifiable. For illustratation consider the total effect $P(y_t \mid x_{t-1})$ and the SCG in Figure 1i where $W$ satisfies the standard front-door criterion with respect to $X$ and $Y$. However, the total effect is not identifiable using $W$ because in the FT-ADMG in Figure 1g or Figure 1h (which is assumed to be unknown), there is a back-door path between $X_{t-1}$ and $W_t$ passing by $X_t$ which should not be blocked by $X_t$ (since $X_t$ is a descendant of $X_{t-1}$ and ancestor of $W_t$). Similarly, the Granger front-door criterion would also incorrectly conclude that the total effect is identifiable. Recall that the Granger front-door criterion assumes the absence of latent confounding across

different time points within a single time series, and explicitly assumes no instantaneous causal relationships and both of these assumptions are violated in the ADMGs shown in Figure1g and Figure 1h. Therefore, it is not surprising that, when relying solely on the SCG and lacking information about the corresponding FT-ADMG, the Granger front-door criterion would mistakenly suggest identifiability of the total effect.

Let us also consider another similar example, with a similar SCG but where $X \to X$ is not in the SCG but there is a larger cycle containing $X$, like the SCG in Figure 3b and suppose that our aim is to identify the total effect $\Pr(y_t \mid do(X_{t-1}))$ with $\gamma_{\max} = 1$. It is possible to construct an FT-ADMG compatible with this SCG in which a back-door path exists: $X_{t-1} \leftarrow U_{t-1} \to X_t \to W_t$. This path cannot be blocked by conditioning on $X_t$ because the presence of a self loop on $X$ implies that $X_t$ could be a descendant of $X_{t-1}$. At the same time, blocking this path via $U_{t-1}$ is also problematic, because it is possible to imagine an alternative FT-ADMG, also compatible with the given SCG, where the same path is instead a directed path: $X_{t-1} \to U_{t-1} \to X_t \to W_t$, implying that $U_{t-1}$ is a descendant of $X_{t-1}$. In this case, conditioning on $U_{t-1}$ would block a valid directed path, further complicating identification. However, in such cases, applying the standard front-door criterion or the Granger front-door criterion would incorrectly suggest that the total effect is identifiable—when, in reality, it is not. Again, this is unsurprising, as these tools were not designed for the specific setting considered in this paper.

At an intuitive level, the failure of the standard front-door criterion and of the Granger front-door criterion in these cases arises due to the simultaneous presence of instantaneous relations between time series and either $X \leftarrow\!\cdots\!\to X$ and a self loop on $X$ in $\mathcal{G}^s$ or just simply larger cycles on $X$. The existence of $X \leftarrow\!\cdots\!\to X$ in $\mathcal{G}^s$ implies that there may be a back-door path between $X_{t-\gamma}$ (for $\gamma > 0$) and $W_t$, starting with $X_{t-\gamma} \leftarrow\!\cdots\!\to X_t$. Meanwhile, the presence of a self loop on $X$ means that this back-door path cannot be blocked using $X_t$, since $X_t$ would be a descendant of $X_{t-\gamma}$. As a result, conditioning on $X_t$ would block a directed path from $X_{t-\gamma}$ to $W_t$, leading to bias in the estimation of the total effect of $X_{t-\gamma}$ to $W_t$. It is important to note that this kind of back-door paths is not immediately apparent when examining only the SCG with classical tools built for DAGs or ADMGs such as the standard front-door criterion. The same intuition holds for the case when there is larger cycles containing $X$ even in the absence of $X \leftarrow\!\cdots\!\to X$ in the SCG.

In other cases, the unsuitability of the standard front-door criterion when applied to SCGs might imply that, even if the total effect of interest is identifiable, the corresponding do-free formula could be more complex than the one associated with the standard front-door criterion (Pearl, 1995) or the Granger front-door criterion (Eichler & Didelez, 2010). For example, consider the total effect $P(y_t \mid x_{t-1})$ and the SCG in Figure 1c (no cycle containing $X$ in the SCG but $X \leftarrow\!\cdots\!\to X$ is in the SCG), where $W$ satisfies the standard front-door criterion with respect to $X$ and $Y$. Now, consider one of the FT-ADMGs given in Figures 1a and 1b compatible with this SCG. In this FT-ADMG, the micro vertices (i.e., $\{W_{t-1}, W_t\}$) corresponding to the macro vertex $W$, which intercepts all directed paths from $X_{t-1}$ to $Y_t$, do not satisfy the standard front-door criterion for $(X_{t-1}, Y_t)$, since there exists a back-door path $X_{t-1} \leftarrow\!\cdots\!\to X_t \to W_t$. However, since there is no cycle containing $X$ in the SCG, this back-door path can be easily blocked by adjusting for $X_t$. Doing so results in a do-free formula similar to the one derived using the front-door criterion, but with the additional step of adjusting for $X_t$. These types of do-free formulas are derived by combining the back-door criterion with the front-door criterion, as discussed in (Pearl, 1995) and subsequently applied by other authors (Fulcher et al., 2019). However, it is important to note that the same do-free formula should not be used if the SCG in Figure 1f (there is cycle containing $X$ in the SCG but $X \leftarrow\!\cdots\!\to X$ is not in the SCG) is considered. Moreover, distinguishing between these different scenarios is not feasible when relying solely on the SCG, whether using classical tools developed for DAGs or ADMGs, such as the standard front-door criterion, or tools tailored to SCGs that assume no latent confounding across time and no instantaneous relations, such as the Granger front-door criterion. These types of unsuitability also exists when considering instantaneous total effects. For instance, take the SCG in Figure 2b and the total effect of $X_t$ on $Y_t$. In this case, there are no back-door paths between $X$ and $W$ within the SCG. However, it is possible to construct an FT-ADMG where a back-door path exists: $X_t \leftarrow U_t \leftarrow \ldots \leftarrow U_{t-1000} \to X_{t-1000} \to W_{t-1000} \to \ldots \to W_t$ (assuming $t_0 < t - 1000$). This path must be blocked to ensure correct identification using the front-door formula. To block this and similar paths, one would need to either adjust on all variables between $X_{t_0}$ and $X_{t-1}$ (which is impractical) or simply adjust on $U_t, U_{t-1}, \ldots, U_{t-\gamma_{\max}}$. However, the standard front-door criterion and its associated theorem cannot be applied to infer this requirement.

## 4 The identifiability of total effects in SCGs with latent confounding

This section, presents the main results of the paper, with some of the corresponding proofs omitted and provided in the appendix. Let us start by giving the extension of the front-door criterion for SCGs.

**Definition 7** (SCG-front-door criterion). *Consider an SCG $\mathcal{G}^s$. A set of macro vertices $\mathbb{W}$ in $\mathcal{G}^s$ satisfy the SCG-front-door criterion relative to a pair of micro vertices $(X_{t-\gamma}, Y_t)$ compatible with a pair of macro vertices $(X, Y)$ in $\mathcal{G}^s$ if:*

1. *$\mathbb{W}$ intercepts all activated directed paths from $X$ to $Y$;*
2. *there is no activated back-door path from $X$ to $\mathbb{W}$;*
3. *all back-door paths from $\mathbb{W}$ to $Y$ are blocked by $X$;*
4. *one of the following holds:*
   - *(a) $Cycles(X, \mathcal{G}^s) = \emptyset$ ; or*
   - *(b) $Cycles(X, \mathcal{G}^s) = \{X \rightarrow X\}$ and $\nexists Z \in Anc(X, \mathcal{G}^s)$ such that $X \leftarrow\!\dashrightarrow Z$ in $\mathcal{G}^s$; or*
   - *(c) $\gamma = 0$.*

Conditions 1-3 in Definition 7 correspond to the three conditions in the standard front-door criterion (Pearl, 1995). For an illustration of Definition 7, Figure 2 provides several examples of SCGs, including the SCGs given in Figures 1f and Figures 1i, where Definition 7 is satisfied for $W$ relative to $(X_{t-\gamma}, Y_t)$. Notice that Definition 7 remains satisfied for $W$ relative to $(X_t, Y_t)$ if a cycle is added on $X$ that does not involve any other vertex in the presented SCGs, as illustrated in Figure 3 which includes the SCG given in Figure 1c. Note that in these SCGs, $W$ is not necessarily the only vertex satisfying Definition 7, for example, in Figure 2e, $U$, also satisfied the SCG-front-door criterion. In contrast, Figure 4 provides several examples of SCGs where Definition 7 is not satisfied for any vertex relative to $(X_{t-\gamma}, Y_t)$.

In the following, lemmas that form the building blocks for the theorem introduced at the end of the section is presented. The first lemma asserts that if a set of macro vertices intercepts all directed paths between $X$ and $Y$ in an SCG, then there exists a finite set of micro vertices that intercepts all directed paths between $X_{t-\gamma}$ and $Y_t$ in any candidate FT-ADMG. This mirrors the first Condition of the standard front-door criterion for ADMGs.

**Lemma 4.1.** *Consider an SCG $\mathcal{G}^s$. If a set of macro vertices $\mathbb{W}$ intercepts all directed paths from $X$ to $Y$ in $\mathcal{G}^s$ then $\{(\mathbb{W}_{t-\gamma+\ell})_{0 \leq \ell \leq \gamma}\}$ intercepts all directed paths from $X_{t-\gamma}$ to $Y_t$ in any candidate FT-ADMG in $\mathcal{C}(\mathcal{G}^s)$.*

For an illustration of this lemma, consider any FT-ADMG depicted in Figure 1 or Figure 5, where in the corresponding SCG, all directed paths from $X$ to $Y$ are intercepted by $W$. Observe that any directed path from $X_{t-1}$ to $Y_t$ must pass through a micro vertex $W_{t-\lambda}$ that corresponds to $W$. If $\lambda > \gamma$, then all active paths from $X_{t-1}$ to $Y_t$ passing through $W_{t-\lambda}$ are not directed paths since $W_{t-\lambda}$ is temporally prior to $t$ and $t - \gamma$. Conversely, if $\lambda < 0$, all paths from $X_{t-1}$ to $Y_t$ that pass through $W_{t-\lambda}$ are blocked, as $W_{t-\lambda}$ would always act as a collider due to its temporal positioning after both $t$ and $t - \gamma$.

The second lemma asserts that given Conditions 1-3, along with Condition 4a, it is guaranteed that there exists a finite set that blocks all back-door paths from $X_{t-\gamma}$ to any vertex in any set of micro vertices that intercepts all directed paths from $X_{t-\gamma}$ to $Y_t$ in any candidate FT-ADMG. Furthermore, this finite set does not contain any descendants of $X_{t-\gamma}$. This mirrors the second Condition of the standard front-door criterion for ADMGs.

**Lemma 4.2.** *Consider an SCG $\mathcal{G}^s = (\mathbb{S}, \mathbb{E}^s)$ and the pair of micro vertices $(X_{t-\gamma}, Y_t)$ compatible with the macro vertices $(X, Y)$. Suppose $\mathbb{W}$ is a set of macro vertices that statisfies Conditions 1, 2 and 3 of Definition 7 in $\mathcal{G}^s$ relative to the pair of micro vertices $(X_{t-\gamma}, Y_t)$. If $Cycles(X, \mathcal{G}^s) = \emptyset$ then for any $W_{t-\lambda} \in \{(\mathbb{W}_{t-\gamma+\ell})_{0 \leq \ell \leq \gamma}\}$, the set $\{(B_{t-\gamma-\ell})_{0 \leq \ell \leq \gamma_{\max}} | B \in Par(X, \mathcal{G}^s)\} \cup \{(B_{t-\gamma-\ell})_{1 \leq \ell \leq \gamma_{\max}} | B \in Anc(\mathbb{W}, \mathcal{G}^s) \cap Desc(X, \mathcal{G}^s)\} \cup \{(X_{t-\gamma+\ell})_{1 \leq \ell \leq \gamma}\}$ blocks all back-door paths from $X_{t-\gamma}$ to $W_{t-\lambda}$ and does not contain any descendant of $X_{t-\gamma}$ in any candidate FT-ADMG in $\mathcal{C}(\mathcal{G}^s)$.*

The intuition behind Lemma 4.2 is that conditioning on micro variables corresponding to the parents of $X$ in the SCG blocks all back-door paths that start with $X_{t-\gamma} \leftarrow$. This still leaves some possible back-door paths

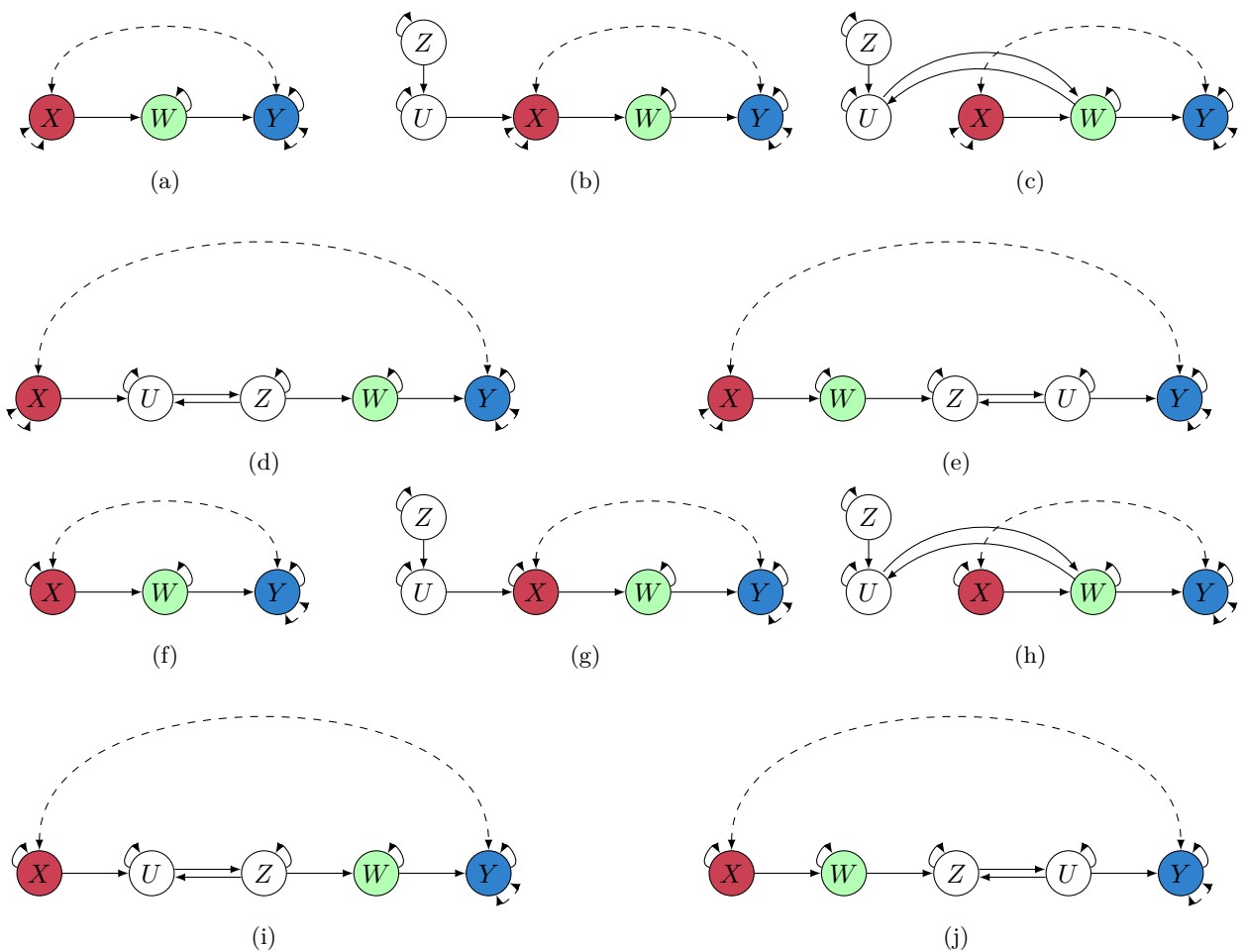

Figure 2: Ten SCGs satisfying Definition 7 for $W$ relative to the pair of micro vertices $(X_{t-\gamma}, Y_t)$, $\forall \gamma \in \{0, \cdots, \gamma_{\max}\}$, i.e., for the first half of these SCGs, Conditions 1, Conditions 2, Conditions 3, and Conditions 4a in Definition 7 are satisfied and for the second half, Conditions 1, Conditions 2, Conditions 3, and Conditions 4b in Definition 7 are satisfied. Each pair of red and blue vertices represents the total effect of interest and green vertices are those that intercept all directed paths from the cause to the effect of interest.

active, those starting with $X_{t-\gamma} \dashleftarrow\dashrightarrow$, which require additional conditioning sets to be blocked, namely some micro variables corresponding to the ancestors of $\mathbb{W}$. The set $\{X_{t-2}, X_t\}$ satisfies the conditions of this lemma when considering the total effect of $X_{t-1}$ on $Y_t$ and the SCG presented in Figure 1c. Specifically, $\{X_{t-2}, X_t\}$ effectively blocks all back-door paths from $X_{t-1}$ to $\{W_{t-1}, W_t\}$ and does not include any descendant of $X_{t-1}$ in any FT-ADMG that is compatible with the SCG (e.g., Figure 1a, Figure 1b). For another, more complete, visual explanation of why the selected set in Lemma 4.2 blocks all back-door paths, refer to the FT-ADMG in Figure 5a. In the figure, the vertices corresponding to $\{(B_{t-\gamma-\ell})_{0 \leq \ell \leq \gamma_{\max}} | B \in Par(X, \mathcal{G}^s)\}$ are highlighted in brown, the vertices corresponding to $\{(B_{t-\gamma-\ell})_{1 \leq \ell \leq \gamma_{\max}} | B \in Anc(\mathbb{W}, \mathcal{G}^s) \cap Desc(X, \mathcal{G}^s)\}$ are highlighted in pink, and the vertices corresponding to $\{(X_{t-\gamma+\ell})_{1 \leq \ell \leq \gamma}\}$ are highlighted in gray.

Notice that if $Cycle(X, \mathcal{G}^s) \neq \emptyset$, then there could exist an FT-ADMG in $\mathcal{C}(\mathcal{G}^s)$ where micro vertices related to $X$ that temporally succeed $X_{t-\gamma}$ can be descendants of $X_{t-\gamma}$. Additionally, these descendants can share a latent confounder with $X_{t-\gamma}$, and the path responsible of this latent confounding cannot be blocked. For example, in the FT-ADMG in Figure 1g compatible with the SCG in Figure 1i that contain a cycle of size 1 on $X$, it is clear that $X_t$ is a descendant of $X_{t-1}$ and that the back-door path from $X_{t-1}$ to $W_t$ starting with $X_{t-1} \dashleftarrow\dashrightarrow X_t$ cannot be blocked by any vertex. Thus, $\{W_{t-1}, W_t\}$ cannot be used to intercept the relationship between $X_{t-1}$ and $Y_t$ for identification using the front-door criterion. However, it turns out this

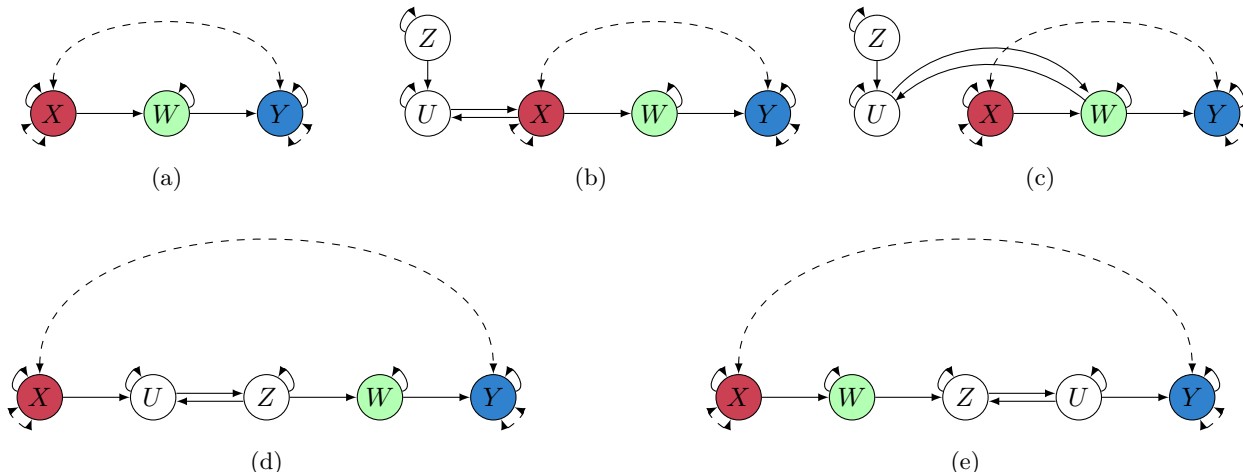

Figure 3: Five SCGs satisfying Definition 7 for $W$ relative only to the pair of micro vertices $(X_t, Y_t)$, i.e., Conditions 1, Conditions 2, Conditions 3, and Conditions 4c in Definition 7 are satisfied. Each pair of red and blue vertices represents the total effect of interest and green vertices are those that intercept all directed paths from the cause to the effect of interest.

is not the case when either the only cycle on $X$ is a self loop and there is no latent confounding between $X$ and its ancestors—which implies that $X \dashleftarrow\dashrightarrow X$ is not in $\mathcal{G}^s$ (as in the SCG in Figure 1f)—or when $\gamma = 0$, as stated in the following two lemmas.

**Lemma 4.3.** *Consider an SCG $\mathcal{G}^s = (\mathbb{S}, \mathbb{E}^s)$ and the pair of micro vertices $(X_{t-\gamma}, Y_t)$ compatible with the macro vertices $(X, Y)$. Suppose $\mathbb{W}$ is a set of macro vertices that statisfies Conditions 1, 2 and 3 of Definition 7 in $\mathcal{G}^s$ relative to the pair of micro vertices $(X_{t-\gamma}, Y_t)$. If $Cycle(X, \mathcal{G}^s) = \{X \to X\}$ and $/\exists Z \in Anc(X, \mathcal{G}^s)$ such that $X \dashleftarrow\dashrightarrow Z$ in $\mathcal{G}^s$ then for any $W_{t-\lambda} \in \{(\mathbb{W}_{t-\gamma+\ell})_{0 \leq \ell \leq \gamma}\}$, the set $\{(B_{t-\gamma-\ell})_{0 \leq \ell \leq \gamma_{\max}} | B \in Par(X, \mathcal{G}^s)\} \cup \{(X_{t-\gamma-\ell})_{1 \leq \ell \leq \gamma_{\max}}\}$ blocks all back-door paths from $X_{t-\gamma}$ to $W_{t-\lambda}$ and does not contain any descendant of $X_{t-\gamma}$ in any candidate FT-ADMG in $\mathcal{C}(\mathcal{G}^s)$.*

The intuition behind Lemma 4.3 is that, similarly to the previous lemma, conditioning on micro variables (for $t-\gamma$ to $t-\gamma-\gamma_{max}$) corresponding to the parents of $X$ in the SCG blocks all back-door paths that start with $X_{t-\gamma} \leftarrow U_{t-\lambda'}$ where $X \neq U$. This leaves only possible back-door paths starting with $X_{t-\gamma} \leftarrow X_{t-\lambda'}$, which can be blocked by timepoints of $X$ that preceeds $t-\gamma$. The set $\{U_{t-2}, U_{t-1}, X_{t-2}\}$ statisfies Lemma 4.3 for the total effect of $X_{t-1}$ on $Y_t$ when considering the SCG in Figure 1f. Which means the set $\{U_{t-2}, U_{t-1}, X_{t-2}\}$ blocks all the back-door paths between $X_{t-1}$ and $W_t$ in any FT-ADMG compatible with the SCG considered. To visually see why the selected set in Lemma 4.3 blocks all back-door paths, refer to the FT-ADMG in Figure 5b. In the figure, the vertices corresponding to $\{(B_{t-\ell})_{0 \leq \ell \leq \gamma_{\max}} | B \in Par(X, \mathcal{G}^s)\}$ are highlighted in brown and the vertices corresponding to $\{(X_{t-\gamma-\ell})_{1 \leq \ell \leq \gamma_{\max}}\}$ are highlighted in pink.

Compared to Lemma 4.2, in Lemma 4.3, the set $(X_{t-\gamma+\ell}) 1 \leq \ell \leq \gamma$ is not included in the conditioning set for two main reasons: (1) in the context of Lemma 4.3, where self loops are allowed, the temporal variables in this set may be descendants of $Xt-\gamma$, and (2) in this scenario, no back-door paths start with $\dashleftarrow\dashrightarrow$. For the same reason as (2), the only subset considered from $(B_{t-\gamma-\ell}) 1 \leq \ell \leq \gamma\max | B \in Anc(\mathbb{W}, \mathcal{G}^s) \cap Desc(X, \mathcal{G}^s)$ is $(X_{t-\gamma-\ell}) 1 \leq \ell \leq \gamma\max$.

Remark that Lemma 4.3 would fail if larger cycles involving $X$ were considered, meaning cycles on $X$ beyond a simple self loop as discussed in Section 3.

Let us now turn to a lemma, which permits even larger cycles and allows confounding between $X$ and its ancestors, but imposes the restriction that $\gamma$ must be zero.

**Lemma 4.4.** *Consider an SCG $\mathcal{G}^s = (\mathbb{S}, \mathbb{E}^s)$ and the pair of micro vertices $(X_t, Y_t)$ compatible with the macro vertices $(X, Y)$. Suppose $\mathbb{W}$ is a set of macro vertices that statisfies Conditions 1, 2 and 3 of Definition 7 in $\mathcal{G}^s$ relative to the pair of micro vertices*

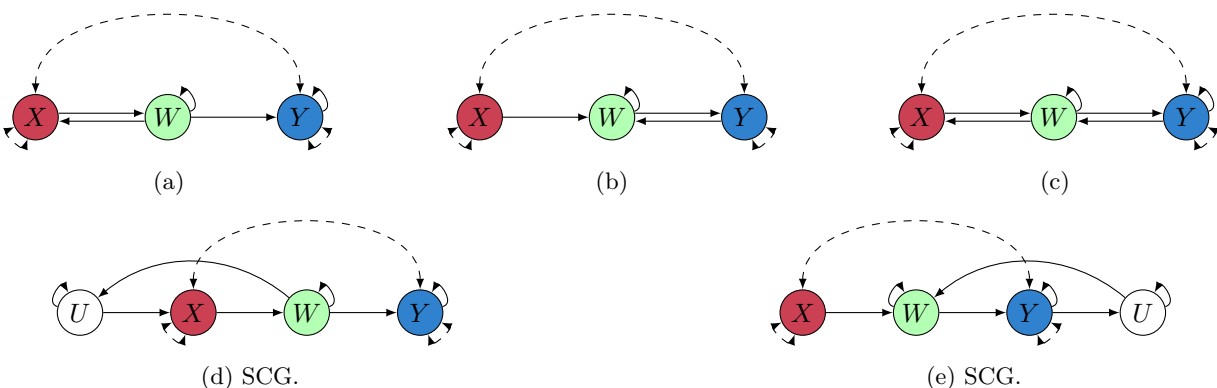

Figure 4: Five SCGs not satisfying Definition 7 for any vertex relative to the micro of vertices $(X_{t-\gamma}, Y_t)$. The SCG in (a) and (d) does not satisfy Definition 7 because Condition 2 is not satisfied. The SCG in (b) and (e) does not satisfy Definition 7 because Condition 3 is not satisfied. The SCG in (c) does not satisfy Definition 7 because Condition 2 and 3 are not satisfied. Each pair of red and blue vertices represents the total effect of interest and green vertices are those that intercepts all directed paths from the cause to the effect of interest.

$(X_t, Y_t)$. *Then for any* $W_t \in \mathbb{W}_t$, *the set* $\{(B_{t-\ell})_{0 \leq \ell \leq \gamma_{\max}} | B \in Anc(X, \mathcal{G}^s) \backslash Desc(X, \mathcal{G})\} \cup$ $\{(B_{t-\ell})_{1 \leq \ell \leq \gamma_{\max}} | B \in (Anc(X, \mathcal{G}^s) \cup Anc(\mathbb{W}, \mathcal{G}^s)) \cap Desc(X, \mathcal{G}^s)\}$ *blocks all back-door paths from* $X_t$ *to* $W_t$ *and does not contain any descendant of* $X_t$ *in any candidate FT-ADMG in* $\mathcal{C}(\mathcal{G}^s)$,

The intuition behind Lemma 4.4 is that conditioning on micro variables corresponding to the parents of $X$ in the SCG cannot block all back-door paths because a parent can also be a descendant of $X$ (since cycles are allowed in this case), but the set micro variables (for $t$ to $t - \gamma_{max}$) corresponding to the ancestors of $X$ that are not descendants of $X$ in the SCG and the set micro variable (for $t - 1$ to $t - \gamma_{max}$) (a subset of the set in purple) corresponding to the ancestors of $X$ that are descendants of $X$ in the SCG blocks all back-door paths that start with $\rightarrow$. This still leaves some possible back-door paths active, those starting with $\leftarrow\text{-}\text{-}\rightarrow$, which require additional conditioning sets to be blocked, namely some of the ancestors of $W$. The set $\{X_{t-1}, W_{t-1}\}$ statisfies Lemma 4.4 for the total effect of $X_t$ on $Y_t$ when considering any of the SCGs given in Figures 1c, 1f and 1i. Which means the set $\{X\}$ blocks all the back-door paths between the $X_t$ and $W_t$ in any FT-ADMG compatible with the SCG considered. To visually see, on a more complete example, why the selected set in Lemma 4.4 blocks all back-door paths, refer to the FT-ADMG in Figure 5c. In the figure, the vertices corresponding to $\{(B_{t-\ell})_{0 \leq \ell \leq \gamma_{\max}} | B \in Anc(X, \mathcal{G}^s) \backslash Desc(X, \mathcal{G})\}$ are highlighted in brown and the vertices corresponding to $\{(B_{t-\ell})_{1 \leq \ell \leq \gamma_{\max}} | B \in (Anc(X, \mathcal{G}^s) \cup Anc(W, \mathcal{G}^s)) \cap Desc(X, \mathcal{G})\}$ are highlighted in pink.

Notice that, compared to Lemma 4.2, in Lemmas 4.3 and 4.4, two key replacements are made regarding the selected conditioning sets: $\{(B_{t-\ell})_{0 \leq \ell \leq \gamma_{\max}} | B \in Par(X, \mathcal{G}^s)\}$ is replaced with $\{(B_{t-\ell})_{0 \leq \ell \leq \gamma_{\max}} | B \in Anc(X, \mathcal{G}^s) \backslash Desc(X, \mathcal{G})\}$ and $\{(B_{t-\gamma-\ell})_{1 \leq \ell \leq \gamma_{\max}} | B \in Anc(\mathbb{W}, \mathcal{G}^s) \cap Desc(X, \mathcal{G}^s)\}$ is replaced with $\{(B_{t-\ell})_{1 \leq \ell \leq \gamma_{\max}} | B \in (Anc(X, \mathcal{G}^s) \cup Anc(\mathbb{W}, \mathcal{G}^s)) \cap Desc(X, \mathcal{G})\}$ to account for potential cycles involving $X$. For example, in the FT-ADMG shown in Figure 5c, conditioning only on the parents, specifically $U_t$ and $U_{t-1}$ (without conditioning on $Z_t$ and $Z_{t-1}$), would be sufficient to block all back-door paths. However, there exists another FT-ADMG compatible with the SCG in Figure 3b, which is almost identical to the FT-ADMG in Figure 5c except that now $\forall t \in [t_0, t_{max}], X_t \rightarrow U_t$ (which means $U_t \not\rightarrow X_t$). In this FT-ADMG, $U_t$ becomes a collider between $X_t$ and $Z_t$, so conditioning on it would activate a new path that cannot be blocked unless the ancestors of $X_t$ are also included in the conditioning set.

The last lemma asserts that given Conditions 1-3, it is guaranteed that there exists a finite set that blocks all back-door paths from some micro vertices (that intercept all directed paths from $X_{t-\gamma}$ to $Y_t$ in any candidate FT-ADMG) to $Y_t$. Furthermore, this finite set does not contain any descendants of the micro vertices that intercept all directed paths. This mirrors the third Condition of the standard front-door criterion for ADMGs.

**Lemma 4.5.** *Consider an SCG $\mathcal{G}^s = (\mathbb{S}, \mathbb{E}^s)$ and the pair of micro vertices $(X_{t-\gamma}, Y_t)$ compatible with the macro vertices $(X, Y)$. Suppose $\mathbb{W}$ is a set of macro vertices that statisfies Conditions 1, 2 and 3 of Definition 7 in $\mathcal{G}^s$ relative to the pair of micro vertices $(X_{t-\gamma}, Y_t)$. Then, for any $W_{t-\lambda} \in \{(\mathbb{W}_{t-\gamma+\ell})_{0 \leq \ell \leq \gamma}\}$, the set $\{(B_{t-\gamma-\ell})_{-\gamma \leq \ell \leq \gamma_{\max}} | B \in Anc(\mathbb{W}, \mathcal{G}^s) \backslash Desc(\mathbb{W}, \mathcal{G}^s)\} \cup \{(B_{t-\gamma-\ell})_{1 \leq \ell \leq \gamma_{\max}} | B \in Anc(\mathbb{W}, \mathcal{G}^s) \cap Desc(\mathbb{W}, \mathcal{G}^s)\}$ blocks all back-door paths from $W_{t-\lambda}$ to $Y_t$ not passing by $\{(\mathbb{W}_{t-\gamma+\ell})_{0 \leq \ell \leq \gamma}\} \backslash \{W_{t-\lambda}\}$ and it does not contain any descendent of $W_{t-\lambda}$.*

The intuition behind Lemma 4.5 is much simpler than the intuition behind the previous lemmas since according to the conditions of Definition 7, there cannot be any back-door paths from $W_{t-\lambda}$ to $Y_t$ starting with $\leftarrow\!\dashrightarrow$ and cycles on $W$ would not create any issues since they either implies active back-door paths that pass by $W_{t-\lambda'}$ such that $\lambda' > \gamma$ which can be easily blocked a set of micro variables, conceptually very similar to the one used in Lemma 4.3 and 4.4. Or they pass by $W_{t-\lambda'}$ such that $W_{t-\lambda'} \in \{(\mathbb{W}_{t-\gamma+\ell})_{0 \leq \ell \leq \gamma}\} \backslash \{W_{t-\lambda}\}$. The set $\{W_{t-2}, X_{t-2}, X_{t-1}, X_t\}$ satisfies the conditions of this lemma when analyzing the total effect of $X_{t-1}$ on $Y_t$ in any of the SCGs presented in Figures 1c, 1f, and 1i. Specifically, $\{W_{t-2}, X_{t-2}, X_{t-1}, X_t\}$ effectively blocks all back-door paths from $W_t$ (respectively, $W_{t-1}$) to $Y_t$ that do not pass through $W_{t-1}$ (respectively, $W_t$). Additionally, this set does not contain any descendant of $W_t$ or $W_{t-1}$ in any FT-ADMG that is compatible with any of these SCGs (e.g., Figures 1a, 1b, 1d, 1e, 1g, or 1h). To gain a clearer visual understanding, using a more complete example, why the selected set in Lemma 4.5 blocks all back-door paths not passing by $\{(\mathbb{W}_{t-\gamma+\ell})_{0 \leq \ell \leq \gamma}\} \backslash \{W_{t-\lambda}\}$, refer to the FT-ADMs in Figure 5d. In these FT-ADMGs, the vertices corresponding to $\{(B_{t-\gamma-\ell})_{-\gamma \leq \ell \leq \gamma_{\max}} | B \in Anc(\mathbb{W}, \mathcal{G}^s) \backslash Desc(\mathbb{W}, \mathcal{G}^s)\}$ are highlighted in orange, the vertices corresponding to $\{(B_{t-\gamma-\ell})_{1 \leq \ell \leq \gamma_{\max}} | B \in Anc(\mathbb{W}, \mathcal{G}^s) \cap Desc(\mathbb{W}, \mathcal{G}^s)\}$ are highlighted in purple, and the vertices corresponding to $\{(\mathbb{W}_{t-\gamma+\ell})_{0 \leq \ell \leq \gamma}\}$ are highlighted in green. The vertex $X_{t-\gamma}$ is highlighted in red and in orange since it is the main cause of interest and at the same time it is in the set $\{(B_{t-\gamma-\ell})_{-\gamma \leq \ell \leq \gamma_{\max}} | B \in Anc(W, \mathcal{G}^s) \backslash Desc(W, \mathcal{G}^s)\}$.

In the following, a sound theorem is introduced for identifying total effects using SCG, even in the presence of cycles and latent confounders.

**Theorem 1.** *If a set of vertices $\mathbb{W}$ satisfies the SCG-front-door criterion (i.e., Definition 7) relative to $(X_{t-\gamma}, Y_t)$ and if $\Pr(\mathbb{s}) > 0$ then $\Pr(y_t \mid do(x_{t-\gamma}))$ is identifiable and is given by the do-free formula:*

$$\Pr(y_t \mid do(x_{t-\gamma})) = \sum_{\mathbb{f}} \sum_{\mathbb{b}^x} \quad \Pr(\mathbb{f} \mid x_{t-\gamma}, \mathbb{b}^x) \Pr(\mathbb{b}^x) \tag{1}$$
$$\times \sum_{\mathbb{b}^f, x'_{t-\gamma}} \Pr(y_t \mid \mathbb{f}, \mathbb{b}^f, x'_{t-\gamma}) \Pr(\mathbb{b}^f, x'_{t-\gamma})$$

*where $\mathbb{F} = \{(\mathbb{W}_{t-\gamma+\ell})_{0 \leq \ell \leq \gamma}\}$;*

$$\mathbb{B}^x = \{(B_{t-\gamma-\ell})_{0 \leq \ell \leq \gamma_{\max}} | B \in Anc(X, \mathcal{G}^s) \backslash Desc(X, \mathcal{G})\}$$
$$\cup \{(B_{t-\gamma-\ell})_{1 \leq \ell \leq \gamma_{\max}} | B \in (Anc(X, \mathcal{G}^s) \cup Anc(\mathbb{W}, \mathcal{G}^s)) \cap Desc(X, \mathcal{G})\} \cup \mathbb{B}^c$$
$$\text{such that } \mathbb{B}^c = \{(X_{t-\gamma+\ell})_{1 \leq \ell \leq \gamma}\} \text{if Condition 4a is satsified, and } \mathbb{B}^c = \emptyset \text{ otherwise;}$$

*and*

$$\mathbb{B}^f = \{(B_{t-\gamma-\ell})_{-\gamma \leq \ell \leq \gamma_{\max}} | B \in Anc(\mathbb{W}, \mathcal{G}^s) \backslash Desc(\mathbb{W}, \mathcal{G}^s))\}$$
$$\cup \{(B_{t-\gamma-\ell})_{1 \leq \ell \leq \gamma_{\max}} | B \in (Anc(\mathbb{W}, \mathcal{G}^s) \cap Desc(\mathbb{W}, \mathcal{G}^s))\} \backslash \{X_{t-\gamma}\}.$$

*Proof.* By Lemma 4.1, since $\mathbb{W}$ intercepts all directed paths from $X$ to $Y$ in the SCG, then $\{(\mathbb{W}_{t-\gamma+\ell})_{0 \leq \ell \leq \gamma}\}$, i.e., $\mathbb{F}$, intercepts all directed paths from $X_{t-\gamma}$ to $Y_t$ in any candidate FT-ADMG in $\mathcal{C}(\mathcal{G}^s)$. Given this result, by the law of total probability, the total effect $\Pr(y_t \mid do(x_{t-\gamma}))$ can be computed in two steps: first, by computing $\Pr(\mathbb{f} \mid do(x_{t-\gamma}))$ and $\Pr(y_t \mid do(\mathbb{f}))$, and subsequently multiplying the two quantities together while summing over $\mathbb{f}$.

Notice that the set used in Lemma 4.2 and the set used in Lemma 4.3 is a subset of $\mathbb{B}^x$ and at the same time $\mathbb{B}^x$ cannot contain any descendant of $X_{t-\gamma}$ in the context of Lemma 4.2 and in the context of Lemma 4.3.

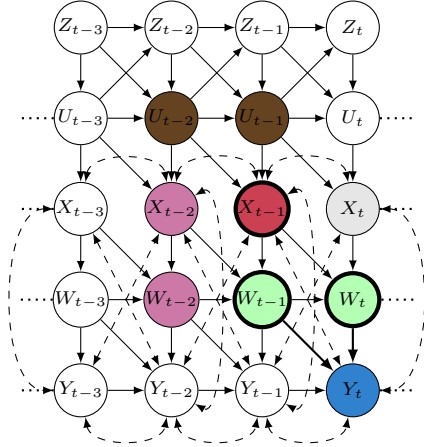

(a) A candidate FT-ADMG of the SCG in Figure 2b showing the set introduced in Lemma 4.2 to block all back-door paths from $X_{t-1}$ to $W_{t-1}$ and $W_t$.

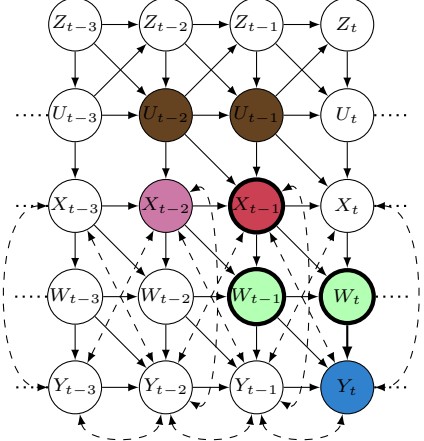

(b) A candidate FT-ADMG of the SCG in Figure 2g showing the set introduced in Lemma 4.3 to block all back-door paths from $X_{t-1}$ to $W_{t-1}$ and $W_t$.

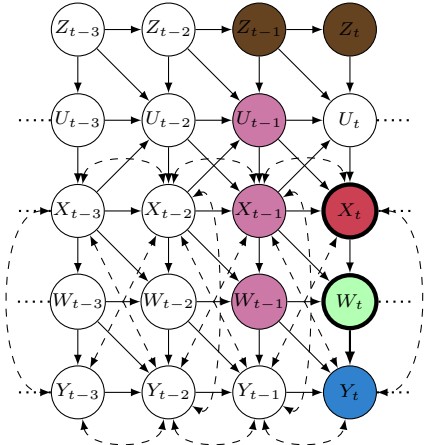

(c) A candidate FT-ADMG of the SCG in Figure 3b showing the set introduced in Lemma 4.4 to block all back-door paths from $X_t$ to $Y_t$.

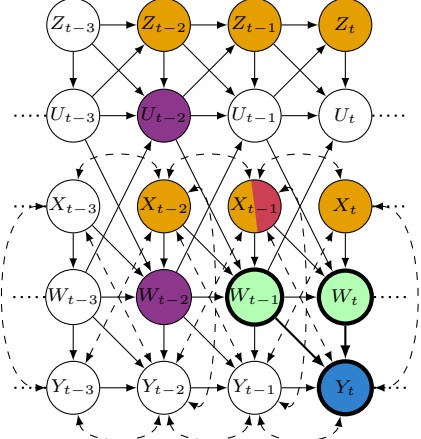

(d) A candidate FT-ADMG of the SCG in Figure 2c showing the set introduced in Lemma 4.5 to block all back-door paths from $W_{t-1}$ and $W_t$ to $Y_t$.

Figure 5: Four FT-ADMGs that correspond respectively to the SCGs in Figures 2b, 2g 3b, and 2c. In each graph, the red and blue vertices represent the total effect of interest, while the green vertices are those that intercept all directed paths from the cause to the effect of interest. In FT-ADMGs (a), (b), and (c), all back-door paths from the red vertex (in bold) to the green vertices (in bold) are blocked by the brown, pink, and gray vertices. In FT-ADMG (d), all back-door paths from each green vertex (in bold)to the blue vertex (in bold) are blocked by the orange, purple, and other green vertices that are temporally prior to the selected green vertex.

Therefore, by Lemma 4.2, 4.3, and 4.4, $\mathbb{B}^x$ statisfies the standard back-door criterion (Pearl, 1995) relative to $(X_{t-\gamma}, \mathbb{F})$ which means that by Pearl (1995, Theorem 1), $\Pr(\mathbb{F} \mid do(x_{t-\gamma})) = \sum_{\mathbb{b}^x} \Pr(\mathbb{F} \mid x_{t-\gamma}, \mathbb{b}^x) \Pr(\mathbb{b}^x)$.

By Lemma 4.5, $\mathbb{B}^f \cup \{X_{t-\gamma}\}$ blocks all back-door paths from $W_{t-\lambda}$ to $Y_t$ for all $0 \leq \lambda \leq \gamma$, except those passing by $\{(\mathbb{W}_{t-\gamma+\ell})_{0 \leq \ell \leq \gamma}\} \backslash \{W_{t-\lambda}\}$ and it does not contain any descendant of $W_{t-\lambda}$. Therefore, since it is intervened on $\mathbb{F} = \{(\mathbb{W}_{t-\gamma+\ell})_{0 \leq \ell \leq \gamma}\}$, $\mathbb{B}^f \cup \{X_{t-\gamma}\}$ satisfies the back-door criterion relative to $(\mathbb{F}, Y_t)$, which means that by Pearl (1995, Theorem 1), $\Pr(y_t \mid do(\mathbb{F})) = \sum_{\mathbb{b}^f, x'_{t-\gamma}} \Pr(y_t \mid \mathbb{F}, \mathbb{b}^f, x'_{t-\gamma},) \Pr(\mathbb{b}^f)$. $\qquad\square$

For simplicity, Theorem 1 is stated under a strict positivity condition on the distribution, which should ideally be relaxed, as discussed in Hwang et al. (2024). Furthermore, the sets used in Equation 1 of Theorem 1 are applicable whether Condition 4a or Condition 4b or Condition 4c of Definition 7 is statisfied. Nevertheless, by Lemma 4.2 the set $\mathbb{B}^x$ in Theorem 1 can be reduced to $\{(B_{t-\gamma-\ell})_{0 \leq \ell \leq \gamma_{\max}} | B \in Par(X, \mathcal{G}^s)\} \cup \{(B_{t-\gamma-\ell})_{1 \leq \ell \leq \gamma_{\max}} | B \in Anc(\mathbb{W}, \mathcal{G}^s) \cap Desc(X, \mathcal{G}^s)\} \cup \{(X_{t-\gamma+\ell})_{1 \leq \ell \leq \gamma}\}$ when Condition 4a is satisfied and it can be reduced to $\{(B_{t-\gamma-\ell})_{0 \leq \ell \leq \gamma_{\max}} | B \in Par(X, \mathcal{G}^s)\} \cup \{(X_{t-\gamma-\ell})_{1 \leq \ell \leq \gamma_{\max}}\}$ when Condition 4b is satisfied.

Definition 7 reveals a broad class of SCGs that enable the identification of the total effect using the do-free formula provided in Theorem 1. This significantly enhances the applicability of causal inference in real-world scenarios where the fully specified causal graph is unknown, but its SCG is available. For example, consider a conceptual study aimed at evaluating the effect of sedation levels administered today on blood pressure regulation tomorrow. Mathematically, this corresponds to estimating $\Pr(BloodPressure_{tomorrow} = b \mid Sedative_{today} = s)$ for a given patient in critical care. Suppose daily patient monitoring records sedation levels, arterial pressure, and heart rate variability over time. Let us assume that the SCG corresponding to these time series is the one depicted in Figure 1c, where $X$ represents sedation levels, $Y$ corresponds to arterial pressure, and $W$ denotes heart rate variability. In this setup, the effect of sedation on blood pressure is fully mediated by heart rate variability. Note that the SCG encodes the assumption of the presence of a latent confounder (physiological factor) between sedation needs and blood pressure regulation, and latent confounding between different time points of $X$ and latent confounding between different timepoints of $Y$. In this scenario, the total effect of interest is not identifiable by adjustment alone, as the back-door criterion is not applicable. The standard front-door criterion and the Granger front-door criterion are not applicable for the reasons given in Section 3. However, the SCG-front-door criterion and the corresponding theorem provide the appropriate tools to correctly quantify the total effect of sedation levels administered today on blood pressure regulation tomorrow from non-experimental data, assuming all underlying assumptions hold. In this case, the appropriate do-free formula is given by:

$$\Pr(y_t \mid x_{t-1}) = \sum_{w_t, w_{t-1}} \sum_{x_{t-2}, x_t} \Pr(w_t, w_{t-1} \mid x_{t-1}, x_{t-2}, x_t)$$
$$\sum_{w_{t-2}, x_{t-2}, x_t, x'_{t-1}} \Pr(y_t \mid w_t, w_{t-1}, w_{t-2}, x_{t-2}, x_t, x'_{t-1}) \Pr(w_{t-2}, x_{t-2}, x_t, x'_{t-1}).$$

## 5 Towards non identifiability results

This section explains why the total effect $\Pr(y_t \mid do(x_{t-\gamma}))$ is not identifiable in any of the SCGs shown in Figure 4.

Consider any SCG in Figure 4. $\Pr(y_t \mid do(x_{t-\gamma}))$ is not identifiable by the back-door criterion (Pearl, 1995) because the back-door path $X_{t-\gamma} \leftarrow\!\dashrightarrow Y_t$ cannot be blocked by any observed micro vertex. In the following, it is demonstrated that for each SCG, the total effect cannot be identified by decomposing it into other total effects and then multiplying them together.

Consider the SCG $\mathcal{G}^s$ in Figure 4a. Let us show that if $X \leftrightarrows W$ in $\mathcal{G}^s$, then there exists at least one $\lambda$ where $0 \leq \lambda < \gamma_{\max}$ such that $\Pr(w_{t-\lambda} \mid do(x_{t-\gamma}))$ is not identifiable. For $\gamma = \lambda$, there exists an FT-ADMG $\mathcal{G}_1 \in \mathcal{C}(\mathcal{G}^s)$ such that $X_t \in Par(W_t, \mathcal{G}_1)$ and another FT-ADMG $\mathcal{G}_2 \in \mathcal{C}(\mathcal{G}^s)$ where $W_t \in Par(X_t, \mathcal{G}_2)$. Thus, the total effect is not identifiable. The same logic can be applied to the SCGs in Figure 4c and a similar logic for Figure 4b to show that $\Pr(y_t \mid do(w_t))$ is not identifiable.

Consider the SCG $\mathcal{G}^s$ in Figure 4d. Let us demonstrate that if $X \to W \to U \to X$ exists in $\mathcal{G}^s$, then there exists at least one $\lambda$ where $0 \le \lambda < \gamma_{\max}$ such that $\Pr(w_{t-\lambda} \mid do(x_{t-\gamma}))$ is not identifiable. Let's take $\gamma = \lambda$. Since there is a back-door path $\pi^s = \langle X, U, W \rangle$, there exists an FT-ADMG $\mathcal{G}_1$ with the back-door path $\pi_1 = X_t \leftarrow U_t \leftarrow W_t$, which would need to be blocked by conditioning on $U_t$. However, there is also a directed path $\pi^s = \langle X, W, Z \rangle$, which corresponds to an FT-ADMG $\mathcal{G}_2$ with the directed path $\pi_2 = X_t \to W_t \to U_t$, where conditioning on $U_t$ would bias the estimation of the total effect. This ambiguity around $U_t$ implies that the total effect is not identifiable. A similar argument can be made for the SCG in Figure 4e, to show that $\Pr(y_t \mid do(w_t))$ is not identifiable.

# 6 Conclusion

This paper focuses on identifying total effects from summary causal graphs with latent confounding. Definition 7 establishes graphical conditions that are sufficient, under any underlying positive probability distribution, for the identifiability of the total effect. Additionally, in cases where identifiability is established, Theorem 1 provides a do-free formula for estimating the total effect. These results contribute to the ongoing effort to understand and estimate total effects from observational data using summary causal graphs. The main limitation of our result is that, while it is sound, it is not complete for identifying total effects using summary causal graphs. Consequently, future work should aim to develop a *complete* identifiability result that accounts for latent confounding and cycles, along with a corresponding do-free formula that does not require any information about the distribution.

### Acknowledgments

I thank Fabrice Carrat, Nathanael Lapidus, and Benjamin Glemain for several discussions about applications of this work in epidemiology. This work was supported by the CIPHOD project (ANR-23-CPJ1-0212-01).

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

## A   Appendix

**Lemma 4.1.** *Consider an SCG $\mathcal{G}^s$. If a set of macro vertices $\mathbb{W}$ intercepts all directed paths from $X$ to $Y$ in $\mathcal{G}^s$ then $\{(\mathbb{W}_{t-\gamma+\ell})_{0\leq\ell\leq\gamma}\}$ intercepts all directed paths from $X_{t-\gamma}$ to $Y_t$ in any candidate FT-ADMG in $\mathcal{C}(\mathcal{G}^s)$.*

*Proof.* Let $\mathcal{G}$ be a candidate FT-ADMG in $\mathcal{C}(\mathcal{G}^s)$. Consider a directed path $\pi$ from $X_{t-\gamma}$ to $Y_t$ in $\mathcal{G}$. Suppose $\pi$ does not contain any vertex from $\{(\mathbb{W}_{t-\gamma+\ell})_{0\leq\ell\leq\gamma}\}$. Then, $\pi$ must include at least one vertex from $\{(\mathbb{W}_{t-\gamma+\ell})_{\ell\in[t_0,t_{max}]\setminus\{0,\cdots,\gamma\}}\}$, as all paths from $X$ to $Y$ are intercepted by $\mathbb{W}$ in $\mathcal{G}^s$. Consider the case where there exists $W_{t-\gamma+\ell}$ in $\pi$ such that $\ell < 0$. In this case, notice that $W_{t-\gamma+\ell}$ is temporally prior to $X_{t-\gamma}$ which means that by Assumption 1, $\pi$ cannot be a directed path because there exists no directed path from $X_{t-\gamma}$ to $W_{t-\gamma+\ell}$. Consider the case where there exists $W_{t-\gamma+\ell}$ in $\pi$ such that $\ell > \gamma$. In this case, notice that $Y_t$ is temporally prior to $W_{t-\gamma+\ell}$ which means that by Assumption 1, $\pi$ cannot be a directed path because there exists no directed path from $W_{t-\gamma+\ell}$ to $Y_t$. Therefore, it must be the case that $\pi$ includes at least one vertex from $(\mathbb{W}_{t-\gamma+\ell})_{0\leq\ell\leq\gamma}$. Since $\mathcal{G}$ and $\pi$ are arbitrary, this conclusion applies to all directed paths between from $X_{t-\gamma}$ to $Y_t$ in any candidate FT-ADMG in $\mathcal{C}(\mathcal{G}^s)$. $\square$

**Property 1.** *Consider an SCG $\mathcal{G}^s$. If there is no backdoor path from $X$ to $W$ in $\mathcal{G}$, then each active backdoor paths from $X_{t-\gamma}$ to $W_{t-\lambda}$ in any FT-ADMG in $\mathcal{C}(\mathcal{G}^s)$ has to pass by another vertex $X_{t-\lambda'}$ with $\lambda' \neq \gamma$.*

*Proof.* Consider an SCG $\mathcal{G}^s$ and a given candidate FT-ADMG $\mathcal{G}$ in $\mathcal{C}(\mathcal{G}^s)$. First, note that a backdoor path from $X_{t-\gamma}$ to $Y_t$ in $\mathcal{G}$ cannot consist solely of edges pointing toward $X_{t-\gamma}$ because if this were the case, there would be a backdoor from $X$ to $W$ path in $\mathcal{G}^s$. This would contradict the assumption that there is no backdoor path from $X$ to $W$. Therefore, the only possible backdoor path, is a path containing at least one ancestor of $X_{t-\gamma}$ and $W_{t-\lambda}$. Consider an active backdoor path $\pi$ in $\mathcal{G}$: $X_{t-\gamma} \leftarrow U_{t'},\cdots,U_{t''},\cdots \leftarrow S_{t'''}\cdots,Q_{t''''} \rightarrow \cdots,Z_{t'''''},\cdots,Z_{t''''''} \rightarrow W_{t-\lambda}$. Let's assume that there exists no $X_{t-\lambda'}$ such that $\lambda' \neq \gamma$ that belongs to $\pi$. Since the path $\pi$ does not pass through any instance of $X$ other than $X_{t-\gamma}$, the path that is compatible with $\pi$ in $\mathcal{G}^s$ is the following path : $X \leftarrow U \cdots \leftarrow S \cdots Q \rightarrow \cdots, Z \rightarrow W$ where each subpath of micro vertices with the left and right endpoints shares the same macro vertex are replaced by this macro vertex. The path we've identified corresponds directly to a backdoor path in $\mathcal{G}^s$. This contradiction implies that the assumption we took must be false. Moreover, if we assume that $X_{t-\lambda'}$ belongs to the path then if we replace $Q_{t''''}$ by $X_{t-\lambda'}$ then the compatible path in $\mathcal{G}^s$ would be $X \rightarrow Z \rightarrow W$ which is not a backdoor path. $\square$

**Lemma 4.2.** *Consider an SCG $\mathcal{G}^s = (\mathbb{S}, \mathbb{E}^s)$ and the pair of micro vertices $(X_{t-\gamma}, Y_t)$ compatible with the macro vertices $(X, Y)$. Suppose $\mathbb{W}$ is a set of macro vertices that statisfies Conditions 1, 2 and 3 of Definition 7 in $\mathcal{G}^s$ relative to the pair of micro vertices $(X_{t-\gamma}, Y_t)$. If $Cycles(X, \mathcal{G}^s) = \emptyset$ then for any $W_{t-\lambda} \in \{(\mathbb{W}_{t-\gamma+\ell})_{0\leq\ell\leq\gamma}\}$, the set $\{(B_{t-\gamma-\ell})_{0\leq\ell\leq\gamma_{\max}}|B \in Par(X, \mathcal{G}^s)\} \cup \{(B_{t-\gamma-\ell})_{1\leq\ell\leq\gamma_{\max}}|B \in Anc(\mathbb{W}, \mathcal{G}^s) \cap Desc(X, \mathcal{G}^s)\} \cup \{(X_{t-\gamma+\ell})_{1\leq\ell\leq\gamma}\}$ blocks all back-door paths from $X_{t-\gamma}$ to $W_{t-\lambda}$ and does not contain any descendant of $X_{t-\gamma}$ in any candidate FT-ADMG in $\mathcal{C}(\mathcal{G}^s)$.*

*Proof.* By definition of $\gamma_{\max}$, the set $\{(B_{t-\gamma-\ell})_{0\leq\ell\leq\gamma_{\max}}|B \in Par(X, \mathcal{G}^s)\}$ contains all observed parents of $X_{t-\gamma}$ in any FT-ADMG in $\mathcal{C}(\mathcal{G}^s)$. Given that $Cycles(X, \mathcal{G}^s) = \emptyset$, if $B \in Par(X, \mathcal{G}^s)$, then $B \notin Desc(X, \mathcal{G}^s)$, implying that $\{(B_{t-\gamma-\ell})_{0\leq\ell\leq\gamma_{\max}}|B \in Par(X, \mathcal{G}^s)\}$ cannot include any descendant of $X_{t-\gamma}$. Therefore, this set blocks all backdoor paths from $X_{t-\gamma}$ to $W_{t-\lambda}$ that begin with an undashed left arrow, i.e., "$X_{t-\gamma} \leftarrow$" without activating a new path between $X_{t-\gamma}$ and $W_{t-\lambda}$. Again, since $Cycles(X, \mathcal{G}^s) = \emptyset$, the only backdoor paths from $X_{t-\gamma}$ to $W_{t-\lambda}$ that is not blocked or becomes activated by $\{(B_{t-\gamma-\ell})_{0\leq\ell\leq\gamma_{\max}}|B \in Par(X, \mathcal{G}^s)\}$ are those that are those starting with "$X_{t-\gamma} \leftarrow\cdots\rightarrow$". By Property 1, all backdoor paths from $X_{t-\gamma}$ to $W_{t-\lambda}$ in any FT-ADMG $\mathcal{G}$ in $\mathcal{C}(\mathcal{G}^s)$ has to pass by another vertex $X_{t-\lambda'}$ with $\lambda' \neq \gamma$. Let us consider three cases:

Case 1 If $\lambda' < 0$ then obviously, by Assumption 1, the backdoor path is necessarily blocked by the empty set since $X_{t-\lambda'}$ cannot be an ancestor of $W_{t-\lambda}$ nor an ancestor of $X_{t-\gamma}$.

Case 2 If $\lambda' > \gamma$, then by Condition 2, $\{(X_{t-\gamma-\ell})_{\ell\leq 1}\}$ can blocks the backdoor path, however this set is unbounded and therefore unpratical. So instead, we can replace it by $\{(B_{t-\gamma-\ell})_{1\leq\ell\leq\gamma_{\max}}|B \in Anc(\mathbb{W}, \mathcal{G}^s) \cap Desc(X, \mathcal{G}^s)\}$ (remark that $\{(X_{t-\gamma-\ell})_{1\leq\ell\leq\gamma_{\max}}\}$ is included in that set) to blocks the

backdoor path. Furthermore, by Assumption 1, $\{(B_{t-\gamma-\ell})_{1\leq\ell\leq\gamma_{\max}}|B\in Anc(\mathbb{W},\mathcal{G}^s)\cap Desc(X,\mathcal{G}^s)\}$ cannot contain descendants of $X_{t-\gamma}$.

Case 3 If $0\leq\lambda'<\gamma$, then by Condition 2, $\{(X_{t-\gamma+\ell})_{1\leq\ell\leq\gamma}\}$ blocks the backdoor path. Furthermore, since $Cycles(X,\mathcal{G}^s)=0$, $\{(X_{t-\gamma+\ell})_{1\leq\ell\leq\gamma}\}$ cannot contain descendants of $X_{t-\gamma}$. $\qquad\square$

**Lemma 4.3.** *Consider an SCG $\mathcal{G}^s=(\mathbb{S},\mathbb{E}^s)$ and the pair of micro vertices $(X_{t-\gamma},Y_t)$ compatible with the macro vertices $(X,Y)$. Suppose $\mathbb{W}$ is a set of macro vertices that statisfies Conditions 1, 2 and 3 of Definition 7 in $\mathcal{G}^s$ relative to the pair of micro vertices $(X_{t-\gamma},Y_t)$. If $Cycle(X,\mathcal{G}^s)=\{X\to X\}$ and $\not\exists Z\in Anc(X,\mathcal{G}^s)$ such that $X\leftarrow\!\dashrightarrow Z$ in $\mathcal{G}^s$ then for any $W_{t-\lambda}\in\{(\mathbb{W}_{t-\gamma+\ell})_{0\leq\ell\leq\gamma}\}$, the set $\{(B_{t-\gamma-\ell})_{0\leq\ell\leq\gamma_{\max}}|B\in Par(X,\mathcal{G}^s)\}\cup\{(X_{t-\gamma-\ell})_{1\leq\ell\leq\gamma_{\max}}\}$ blocks all back-door paths from $X_{t-\gamma}$ to $W_{t-\lambda}$ and does not contain any descendant of $X_{t-\gamma}$ in any candidate FT-ADMG in $\mathcal{C}(\mathcal{G}^s)$.*

*Proof.* Condition 2 implies that there exists no active backdoor path $\pi^s=\langle V^1=X,\ldots,V^n=W\rangle$ from $X$ to $W$ in $\mathcal{G}^s$ such that $\langle V^2,\ldots,V^{n-1}\rangle\subseteq Desc(X,\mathcal{G}^s)$ which means that there are no cycles that contain $X$ and another vertex on a path from $X$ to $W$. Furthermore, by Property 1, all backdoor paths from $X_{t-\gamma}$ to $W_{t-\lambda}$ in any FT-ADMG $\mathcal{G}$ in $\mathcal{C}(\mathcal{G}^s)$ has to pass by another vertex $X_{t-\lambda'}$ with $\lambda'\neq<\gamma$. Let us consider three cases:

Case 1 If $\lambda'<0$, then obviously, by Assumption 1, the backdoor path is necessarily blocked by the empty set since $X_{t-\lambda'}$ cannot be an ancestor of $W_{t-\lambda}$ nor an ancestor of $X_{t-\gamma}$.

Case 2 If $\lambda'\geq\gamma$ and if the second micro vertex $U_{t-\lambda''}$ on the backdoor path we are considering is compatible with the macro vertex $U\notin Cycles(X,\mathcal{G}^s)$ then the backdoor path can be blocked by $\{(B_{t-\gamma-\ell})_{0\leq\ell\leq\gamma_{\max}}|B\in Par(X,\mathcal{G}^s)\}$.

Case 3 If $\lambda'\geq\gamma$ and if the second micro vertex $U_{t-\lambda''}$ on the backdoor path we are considering is compatible with the macro vertex $U\in Cycles(X,\mathcal{G}^s)$ then, since we consider that $Cycle(X,\mathcal{G}^s)=\{X\to X\}$, $U=X$ and $\lambda'>\gamma$ . Therefore, the backdoor path starts with $X_{t-\gamma}\leftarrow X_{t-\lambda'}$ which means it is obviously blocked by $\{(X_{t-\gamma-\ell})_{1\leq\ell\leq\gamma_{\max}}\}$.

In addition, by construction and by Assumption 1, $\{(B_{t-\ell})_{0\leq\ell\leq\gamma_{\max}}|B\in Par(X,\mathcal{G}^s)\}\cup\{(X_{t-\gamma-\ell})_{1\leq\ell\leq\gamma_{\max}}\}$ cannot contain any descendant of $X_{t-\gamma}$. Which means that $\{(B_{t-\ell})_{0\leq\ell\leq\gamma_{\max}}|B\in Par(X,\mathcal{G}^s)\}\cup\{(X_{t-\gamma-\ell})_{1\leq\ell\leq\gamma_{\max}}\}$ blocks all backdoor paths from $X_{t-\gamma}$ to $W_{t-\lambda}$ starting with an undashed right arrow, i.e, "$X_{t-\gamma}\leftarrow$". Since we consider that $\not\exists Z\in Anc(X,\mathcal{G}^s)$ such that $X\leftarrow\!\dashrightarrow Z$, , no additional back-door paths can exist from $X_{t-\gamma}$ to $W_{t-\lambda}$. $\qquad\square$

**Lemma 4.4.** *Consider an SCG $\mathcal{G}^s=(\mathbb{S},\mathbb{E}^s)$ and the pair of micro vertices $(X_t,Y_t)$ compatible with the macro vertices $(X,Y)$. Suppose $\mathbb{W}$ is a set of macro vertices that statisfies Conditions 1, 2 and 3 of Definition 7 in $\mathcal{G}^s$ relative to the pair of micro vertices $(X_t,Y_t)$. Then for any $W_t\in\mathbb{W}_t$, the set $\{(B_{t-\ell})_{0\leq\ell\leq\gamma_{\max}}|B\in Anc(X,\mathcal{G}^s)\backslash Desc(X,\mathcal{G})\}\cup\{(B_{t-\ell})_{1\leq\ell\leq\gamma_{\max}}|B\in(Anc(X,\mathcal{G}^s)\cup Anc(\mathbb{W},\mathcal{G}^s))\cap Desc(X,\mathcal{G}^s)\}$ blocks all back-door paths from $X_t$ to $W_t$ and does not contain any descendant of $X_t$ in any candidate FT-ADMG in $\mathcal{C}(\mathcal{G}^s)$,*

*Proof.* Condition 2 implies that there exists no active backdoor path $\pi^s=\langle V^1=X,\ldots,V^n=W\rangle$ from $X$ to $W$ in $\mathcal{G}^s$ such that $\langle V^2,\ldots,V^{n-1}\rangle\subseteq Desc(X,\mathcal{G}^s)$ which means that there are no cycles that contain $X$ and another vertex on a path from $X$ to $W$. Furthermore, by Property 1, all backdoor paths from $X_t$ to $W_t$ in any FT-ADMG $\mathcal{G}$ in $\mathcal{C}(\mathcal{G}^s)$ has to pass by another vertex $X_{t-\lambda'}$ with $\lambda'\neq 0$. Let us consider three cases:

Case 1 If $\lambda'<0$, then obviously, by Assumption 1, the backdoor path is necessarily blocked by the empty set since $X_{t-\lambda'}$ cannot be an ancestor of $W_t$ nor an ancestor of $X_t$.

Case 2 If $\lambda'>0$ and if the second micro vertex $U_{t-\lambda}$ on the backdoor path we are considering is compatible with the macro vertex $U\notin Cycles(X,\mathcal{G}^s)$ then the backdoor path can be blocked by $\{(B_{t-\ell})_{0\leq\ell\leq\gamma_{\max}}|B\in Anc(X,\mathcal{G}^s)\backslash Desc(X,\mathcal{G})\}$.

**Case 3** If $\lambda' > 0$ and if the second micro vertex $U_{t-\lambda}$ on the backdoor path we are considering is compatible with the macro vertex $U \in Cycles(X, \mathcal{G}^s)$ then the backdoor path can be blocked by $\{(B_{t-\ell})_{0 \leq \ell \leq \gamma_{\max}} | B \in Anc(X, \mathcal{G}^s) \backslash Desc(X, \mathcal{G})\} \cup \{(B_{t-\ell})_{1 \leq \ell \leq \gamma_{\max}} | B \in Anc(X, \mathcal{G}^s) \cap Desc(X, \mathcal{G})\}$.

In addition, by construction and by Assumption 1, $\{(B_{t-\ell})_{0 \leq \ell \leq \gamma_{\max}} | B \in Anc(X, \mathcal{G}^s) \backslash Desc(X, \mathcal{G})\} \cup \{(B_{t-\ell})_{1 \leq \ell \leq \gamma_{\max}} | B \in Anc(X, \mathcal{G}^s) \cap Desc(X, \mathcal{G})\}$ cannot contain any descendant of $X_t$. Which means that $\{(B_{t-\ell})_{0 \leq \ell \leq \gamma_{\max}} | B \in Anc(X, \mathcal{G}^s) \backslash Desc(X, \mathcal{G})\} \cup \{(B_{t-\ell})_{1 \leq \ell \leq \gamma_{\max}} | B \in Anc(X, \mathcal{G}^s) \cap Desc(X, \mathcal{G})\}$ blocks all backdoor paths from $X_t$ to $W_t$ starting with an undashed right arrow, i.e, "$X_t \leftarrow$". This means that the only backdoor paths from $X_t$ to $W_t$ that are not blocked are the ones statring with "$X_t \leftarrow\!\!\dashrightarrow$". Let us consider two cases:

**Case 1** If $\lambda' < 0$, then obviously, by Assumption 1, the backdoor path is necessarily blocked by the empty set since $X_{t-\lambda'}$ cannot be an ancestor of $W_t$ nor an ancestor of $X_t$.

**Case 2** If $\lambda' > 0$, then by Condition 2, $\{(X_{t-\gamma-\ell})_{\ell \leq 1}\}$ can blocks the backdoor path, however this set is unbounded and therefore unpratical. So instead, we can replace it by $\{(B_{t-\ell})_{1 \leq \ell \leq \gamma_{\max}} | B \in Anc(\mathbb{W}, \mathcal{G}^s) \cap Desc(X, \mathcal{G})\}$. Furthermore, by Assumption 1, $\{(B_{t-\ell})_{1 \leq \ell \leq \gamma_{\max}} | B \in Anc(\mathbb{W}, \mathcal{G}^s) \cap Desc(X, \mathcal{G})\}$ cannot contain any descendant of $X_t$ in any candidate FT-ADMG in $\mathcal{C}(\mathcal{G}^s)$. $\qquad\square$

**Property 2.** *Consider an SCG $\mathcal{G}^s$. Given Condition 3, if all backdoor path from $W$ to $Y$ in $\mathcal{G}$ are blocked by $X$, then each active backdoor paths from $W_{t-\lambda}$ to $Y_t$ in any FT-ADMG in $\mathcal{C}(\mathcal{G}^s)$ has to pass either by a vertex $X_{t-\lambda'}$ or a vertex $W_{t-\lambda''}$ with $\lambda'' \neq \lambda$.*

*Proof.* Consider an SCG $\mathcal{G}^s$ and a given candidate FT-ADMG $\mathcal{G}$ in $\mathcal{C}(\mathcal{G}^s)$. The proof that there might exists an active backdoor path from $W_{t-\lambda}$ to $Y_t$ that does not pass through any $X_{t-\lambda'}$ must pass through a vertex $W_{t-\lambda''}$ follows a similar reasoning to that used in the proof of Property 1. Let's assume there is an active backdoor path between $W_{t-\lambda}$ and $Y_t$ that does not pass through $W_{t-\lambda''}$. If the path between $W_{t-\lambda}$ and $Y_t$ is active and does not pass through $W_{t-\lambda'}$, then this path must be relying on connections in the SCG. Consider such an active backdoor path in $\mathcal{G}$: $W_{t-\lambda} \leftarrow U_{t'}, \cdots, U_{t''}, \cdots \leftarrow S_{t'''} \cdots, Q_{t''''} \rightarrow \cdots, Z_{t'''''}, \cdots, Z_{t''''''} \rightarrow Y_t$. The compatible of the path considered in $\mathcal{G}^s$ is the following path : $W \leftarrow U \cdots \leftarrow S \cdots Q \rightarrow \cdots, Z \rightarrow Y$ where each subpath of micro vertices with the left and right endpoints shares the same macro vertex are replaced by this macro vertex. The identified path corresponds to a backdoor path in $\mathcal{G}^s$. If $X$ is not on this path, it would lead to a contradiction, as in that case, not all backdoor paths would be blocked by $X$. This implies that our initial assumption—that there exists a backdoor path between $W_{t-\lambda}$ and $Y_t$ that does not pass through $W_{t-\lambda''}$—must be false. Consequently, under our assumption, $X$ must be on the path in $\mathcal{G}^s$, and $X_{t-\lambda'}$ must be on the corresponding path in $\mathcal{G}$.

Now, under a similar scenario, let us assume that the backdoor path between $W_{t-\lambda}$ and $Y_t$ does pass through $W_{t-\lambda''}$. For instance, suppose $W_{t-\lambda''} = Q_{t''''}$ on this path. In this case, the compatible path in $\mathcal{G}^s$ would be $W \rightarrow \cdots \rightarrow Z \rightarrow Y$, which is not a backdoor path in $\mathcal{G}^s$. $\qquad\square$

**Lemma 4.5.** *Consider an SCG $\mathcal{G}^s = (\mathbb{S}, \mathbb{E}^s)$ and the pair of micro vertices $(X_{t-\gamma}, Y_t)$ compatible with the macro vertices $(X, Y)$. Suppose $\mathbb{W}$ is a set of macro vertices that statisfies Conditions 1, 2 and 3 of Definition 7 in $\mathcal{G}^s$ relative to the pair of micro vertices $(X_{t-\gamma}, Y_t)$. Then, for any $W_{t-\lambda} \in \{(\mathbb{W}_{t-\gamma+\ell})_{0 \leq \ell \leq \gamma}\}$, the set $\{(B_{t-\gamma-\ell})_{-\gamma \leq \ell \leq \gamma_{\max}} | B \in Anc(\mathbb{W}, \mathcal{G}^s) \backslash Desc(\mathbb{W}, \mathcal{G}^s)\} \cup \{(B_{t-\gamma-\ell})_{1 \leq \ell \leq \gamma_{\max}} | B \in Anc(\mathbb{W}, \mathcal{G}^s) \cap Desc(\mathbb{W}, \mathcal{G}^s)\}$ blocks all back-door paths from $W_{t-\lambda}$ to $Y_t$ not passing by $\{(\mathbb{W}_{t-\gamma+\ell})_{0 \leq \ell \leq \gamma}\} \backslash \{W_{t-\lambda}\}$ and it does not contain any descendent of $W_{t-\lambda}$.*

*Proof.* If there is a cycle that includes $W$ and any other vertex not in $\mathbb{W}$ on a directed path from $X$ to $Y$ that is intercepted by $W$ and that is closer than $W$ to $Y$ on the path, then there exists an active backdoor path from $W$ to $Y$ that does not pass through $X$ (which means cannot be blocked by $X$). By Conditions 2 and 3, this cannot be true. Therefore, it must be the case that all cycles containing $W$ do not include any other vertex on a directed path from $X$ to $Y$ that are neither in $\mathbb{W}$ nor closer to $Y$ on the path. By Property 2, all backdoor path from $W_{t-\lambda}$ to $Y_t$ in any FT-ADMG in $\mathcal{C}(\mathcal{G}^s)$ has to pass by either by $X_{t-\lambda'}$ or by $W_{t-\lambda''}$ with $\lambda'' \neq \lambda$.

Let's consider one of these backdoor paths that pass by a vertex $X_{t-\lambda'}$ but does not pass by any vertex $W_{t-\lambda''}$ such that $\lambda'' \neq \lambda$. By Condition 3, $\{(B_{t-\gamma-\ell})_{-\gamma \leq \ell \leq \gamma_{\max}} | B \in Anc(W, \mathcal{G}^s) \backslash Desc(W, \mathcal{G}^s)\}$ block this path, however it can activate a path of the form $W_{t-\lambda} \leftarrow \ldots X_{t-\lambda'} \longleftrightarrow Y_t$ where $\lambda' > \gamma + \gamma_{\max}$. Since $\lambda \leq \gamma$ and $\lambda' > \gamma + \gamma_{\max}$ then the parents of $W_{t-\lambda}$ which are temporally prior to $t - \lambda$ can block this path. Notice that these parents are in $\{(B_{t-\gamma-\ell})_{1 \leq \ell \leq \gamma_{\max}} | B \in Anc(W, \mathcal{G}^s) \cap Desc(W, \mathcal{G}^s)\}$. Furthermore, by construction and by Assumption 1, $\{(B_{t-\gamma-\ell})_{-\gamma \leq \ell \leq \gamma_{\max}} | B \in Anc(W, \mathcal{G}^s) \backslash Desc(W, \mathcal{G}^s)\} \cup \{(B_{t-\gamma-\ell})_{1 \leq \ell \leq \gamma_{\max}} | B \in Anc(W, \mathcal{G}^s) \cap Desc(W, \mathcal{G}^s)\}$ does not contain any descendant of $W_{t-\lambda}$.

Now let's consider one of the remaining backdoor paths that pass by a vertex $W_{t-\lambda''}$. Let us consider two cases:

Case 1 If $\lambda'' > \gamma$, then obviously the backdoor path can be blocked by $\{(B_{t-\gamma-\ell})_{1 \leq \ell \leq \gamma_{\max}} | B \in Anc(W, \mathcal{G}^s) \cap Desc(W, \mathcal{G}^s)\}$.

Case 2 If $\lambda'' \leq \gamma$ then the backdoor path pass by $\{(\mathbb{W}_{t-\gamma+\ell})_{0 \leq \ell \leq \gamma}\} \backslash \{W_{t-\lambda}\}$ which means it does not need to be blocked.

$\square$

