# OpenReview forum: "Towards identifiability of micro total effects in summary causal graphs with latent confounding:  extension of  the front-door criterion"
_TMLR — Accepted by TMLR_

### Review · Reviewer_k1g5 · 2025-02-04

**Summary Of Contributions:**

This is a theoretical paper, that establishes a sufficient graphical condition on summary causal graph (SCG) with latent confoundings for the identifiability of total causal effect on the full-time ADMG (FT-ADMG) - intuitively, formulating a version of the front-door criterion for SCG. For cases when this condition is satisfied, it provides a do-free formula for the computation of the identifiable causal effect.

**Audience:**

Yes

**Broader Impact Concerns:**

I don't see a clear broader impact or risk of harm from this paper, especially given its theoretical nature

**Claims And Evidence:**

Yes

**Requested Changes:**

The requested changes all reflect the points in the weaknesses section in terms of clarity:

1. An intuitive summary on the conditioning variables in Lemmas 4.2-4.4 / a proof sketch in the main text that carries such intuition.
2. An intuition on why, for $\gamma \neq 0$, removing cycles through $X$ is the right choice for defining the front-door criterion of Def. 7.
3. Explanatory captions on Fig. 4, i.e. how the graphs violate Definition 7.

**Strengths And Weaknesses:**

## Strength

**Contribution.** The paper proposes a valuable contribution, in particular

1. An example demonstrating the deficiencies of the standard front-door criterion applied to SCG for identifiability of causal effects on micro (temporal) variables
2. Sufficient graphical condition on the SCG with latent confoundings for the identifiability of total causal effect on micro variables
3. A do-free formula for the computation of identifiable causal effects.

These contributions are surely of interest to the causality community.

**Presentation.** The paper is well-structured and, in general, clearly written. In particular, I praise the clarity of Sections 2 and 3, which introduce all the required background and the limitations of the *standard* front-door criterion naively applied to SCG. The use of graphs as a visual aid for understanding definitions and theoretical results is very helpful.

## Weaknesses

- The lack of intuition on some important definitions or results of the paper makes understanding Section 4 challenging. In particular
    1. It’s clear by example how the standard front-door criterion on SCG is insufficient for the identifiability of causal effect on micro variables. I don’t understand, at an intuitive level, why removing all cycles — as per condition 4a in Def. 7 — would be the right choice. Could the authors clarify this point?
    2. Lemma 4.2, 4.3, 4.4. lacks an intuitive explanation of why the selected variables block the intended backdoor paths. For example, about Lemma 4.2, without reading the proof I couldn’t understand why the proposed set of variables is the right choice to block the backdoor paths between $X_{t-\gamma}$ and $W_{t-\lambda}$. On the other hand, I think the proof does a good job of providing this intuition: conditioning on the parents of $X$ in the SCG blocks all backdoor paths that start with $X_{t-\gamma} \leftarrow$. This still leaves some possible backdoor paths open, those starting with $X_{t-\gamma} \leftrightarrow$, which require additional conditioning sets to be blocked. An explanation of this kind helps the reader break the overall problem - understanding why the big conditioning set is the right one - into simple subproblems - i.e. what *type* of backdoor paths each of the individual subsets is contributing to block.

    A proof sketch in the main text (or a short discussion after the Lemma) might be the right way to provide this intuition.

    In general, the paper is technically very dense, which is why I believe additional guidance in the reading of Section 4 is beneficial.

- Could the authors clarify why Theorem 1 is not complete, i.e. which conditions of Definition 7 are not necessary for the identifiability?

## Questions

I use this place to ask for generic clarifying questions

- To make sure I understand all relevant graphical notions for SCG:
    1. Is $X \rightarrow X$ a cycle? I’d say no, as there is no direct path, as required by the definition
    2. Is $X \rightarrow Y \rightarrow X$ a cycle? Again, I’d say no, as there is no direct path, as required by the definition
- Fig 4a: why is this not satisfying Definition 7? (Perhaps, this is connected to my doubts above on cycles.) To allow the reader to check understanding, it would be good to have in the figure caption which conditions of Def. 7 are violated by Fig. 4 example.
- Def. 5: what does *uniquely* mean in the definition?

## Miscellanea

- Typos: illusrtation, necessartly (page 8); necesserly (page 15), ..
- Often $\mathcal G$ in place of $\mathcal G^s$, e.g. in Lemma 4.3, Property 1
- In Lemma 4.2: $Cycles (X, \mathcal G^s)$.
- After Def. 6, page 6: “macro” (in purple) → “macro” (in blue) - I think.

---

> ### Author Response · Authors · 2025-02-10
> **Response to Reviewer k1g5  (1/2)**
>
> We thank the Reviewer for constructive and valuable comments. Detailed responses to all the points raised by the Reviewer are reported below. We have also corrected the typos highlighted by the Reviewer.
>
> ## Addressing the comments in Weaknesses and Questions
>
> Here, we address the comments and questions raised in the Weaknesses section and in the Questions section. We begin by restating each question before providing our response.
>
> * Question 1: It’s clear by example how the standard front-door criterion on SCG is insufficient for the identifiability of causal effect on micro variables. I don’t understand, at an intuitive level, why removing all cycles — as per condition 4a in Def. 7 — would be the right choice. Could the authors clarify this point?
>
> **Response**:
> At an intuitive level, the front-door criterion fails in this context because, 1) even if it appears to be satisfied in the SCG, the presence of cycles on X can induce back-door paths between $X_{t-\gamma}$ and  $W_{t:t-\gamma}$ (micro variables corresponding to the set of macro variables $W$ satisfying the front-door criterion); 2) There can be a backdoor path between  $X_{t-\gamma}$ ($\gamma>0$) and  $W_{t:t-\gamma}$ starting with $X_{t-\gamma} \leftrightarrow X_t$ that cannot be blocked using $X_t$ whenever there is a cycle on $X$ because $X_t$ would be a descendant of $X_{t-\gamma}$.
> These back-door paths are not immediately apparent when examining only the SCG. However, for the front-door criterion to be correctly applied, there must be no back-door path between   $X_{t-\gamma}$ and $W_{t:t-\gamma}$. Moreover, blocking these paths is often impossible because conditioning on any variable along the path would introduce additional bias. This occurs because these variables are directly influenced by the the exposure (due to the cycle), which means conditioning on them would simultaneously block a directed path from $X_{t-\gamma}$  to $W_{t:t-\gamma}$, thereby distorting the estimation of the causal effect  from $X_{t-\gamma}$ to $W_{t:t-\gamma}$.
>
> * Question 2: Lemma 4.2, 4.3, 4.4. lacks an intuitive explanation of why the selected variables block the intended backdoor paths. For example, about Lemma 4.2, without reading the proof I couldn’t understand why the proposed set of variables is the right choice to block the backdoor paths between and . On the other hand, I think the proof does a good job of providing this intuition: conditioning on the parents of in the SCG blocks all backdoor paths that start with . This still leaves some possible backdoor paths open, those starting with , which require additional conditioning sets to be blocked. An explanation of this kind helps the reader break the overall problem - understanding why the big conditioning set is the right one - into simple subproblems - i.e. what type of backdoor paths each of the individual subsets is contributing to block.
>
> **Response**:
> We agree that adding the intuition will simplify the reading of the paper.
>
> For the intuition behind lemma 4.2 you are absolutely right:  conditioning on micro variables corresponding to the parents of $X$ in the SCG blocks all backdoor paths that start with $\rightarrow$. This still leaves some possible backdoor paths open, those starting with $\leftrightarrow$, which require additional conditioning sets to be blocked, namely some micro variables corresponding to the the ancestors of $W$ (in purple).
>
> The intuition behind lemma 4.3 is that conditioning on micro variables corresponding to  the parents of in the SCG cannot block all backdoor paths because a parent can also be a descendant (since cycles are allowed in this case), but the set micro variables (for $t$ to $t-\gamma_{max}$) (in brown) corresponding to the ancestors that are not descendants in the SCG  and the set micro variable (for $t-1$ to $t-\gamma_{max}$) (a subset of the set in purple) corresponding to the ancestors that are descendants in the SCG  blocks all backdoor paths that start with $\rightarrow$. This still leaves some possible backdoor paths open, those starting with $\leftrightarrow$, which require additional conditioning sets to be blocked, namely some of the ancestors of $W$  (the other subset of the set in purple) .
>
> The intuition behind lemma 4.4 is much simpler since according to the conditions of Definition 7, there cannot be paths from any $W_t-i$ to $Y_t$ starting with $\leftrightarrow$. So a set of micro variables, very similar to the one used in Lemma 4.3 to block all paths starting with $\rightarrow$, can be used (in orange and pink).

---

> > ### Author Response · Authors · 2025-02-10
> > **Response to Reviewer k1g5 (2/2)**
> >
> > * Question 3: Could the authors clarify why Theorem 1 is not complete, i.e. which conditions of Definition 7 are not necessary for the identifiability?
> >
> > **Response**:
> > Theorem 1 is not complete for the same reason that the front-door criterion is not complete. It is possible to imagine examples where the causal effect is identifiable, yet the front-door criterion is not satisfied.
> > For instance, in the context of DAGs, consider two observed variables and no hidden confounding,  $X,Y$, and consider that  $X\rightarrow Y$.  In this case, if we are interested in the total effect of $X$ on $Y$, this effect is clearly identifiable. However, the front-door criterion does not apply because there is no set of variables that fully mediates the relationship between $X$ and $Y$.
> > Similarly, in the context of SCGs, consider an example where $X\rightarrow Y$ in the SCG. Here, the total effect of $X_{t-1}$ on $Y_t$ is identifiable by adjusting on the past of $X_{t-1}$. However, Definition 7 is not satisfied.
> >
> >
> > * Questions 4: Is X-> X a cycle? I’d say no, as there is no direct path, as required by the definition. Is X-> Y -> X a cycle? Again, I’d say no, as there is no direct path, as required by the definition Do these cases represent cycles?
> >
> > **Response**:
> > We apologize for the lack of clarity in the text. According to our definition, the second case constitutes a cycle because there is a directed path from $X$ to $Y$ and there is the edge $Y \rightarrow X$ in the SCG.  However, we acknowledge that this is not explicitly clear for the first case. To improve clarity, we will revise the definition of a cycle by adding the following:
> > « For an SCG $G^s$ , a directed path from $X$ to $Y$ and the edge $Y \rightarrow X$ form a directed cycle in $G^s$. Additionally, we consider the specific case where a node has a directed edge to itself as a cycle. »
> >
> >
> >
> > * Question 5: Fig 4a: why is this not satisfying Definition 7? (Perhaps, this is connected to my doubts above on cycles.) To allow the reader to check understanding, it would be good to have in the figure caption which conditions of Def. 7 are violated by Fig. 4 example
> >
> > **Response**:
> > We hope that our response to the previous question has clarified Figure 4a. We also agree that it would be helpful to specify in the caption which condition of Definition 7 is violated in Figure 4. Therefore, we will add the following:
> > The SCG in (a) and (d) does not satisfy Definition 7 because Condition 2 and Condition 4a of the Definition are not satisfied. The SCG in (b) and (e) does not satisfy Definition 7 because Condition 3 of the Definition is not satisfied. The SCG in (c) does not satisfy Definition 7 because Condition 2, 3 and Condition 4a of the Definition are not satisfied.
> >
> >
> > * Question 6: Def. 5: what does uniquely mean in the definition?
> >
> > **Response**:
> > In definition 5 uniquely mean that the causal effect is identifiable with the same do-free formula from any distribution compatible with any FT-ADMG compatible with the SCG.
> >
> >
> > ## Requested Changes:
> >
> > In the following, we point out how will we make the changes that you have requested:
> > *  The change regarding this point will be made either by incorporating the paragraph from our response to Question 2 or by including the proofs in the main text, depending on your preference.
> > * We will add the paragraph provided in our response to Question 1 to the main text.
> > * We will include the paragraph from our response to Question 5 in the caption of Figure 4.
> > * Additionally, we will revisit the definition of cycles, as mentioned in our response to Question 4. Finally, if you find it valuable, we will also clarify the meaning of "uniquely" in Definition 5 in the main text, as stated in our response to Question 6.
> >
> >
> > We hope our responses adequately address all concerns, and we look forward to any further feedback.

---

> > > ### Comment · Reviewer_k1g5 · 2025-02-11
> > >
> > > I thank the authors for their response. I still have some doubts I would like to clear out:
> > >
> > > 1. Why is it that $X \dashleftarrow\dashrightarrow W$ is not a backdoor path in the SCG. This question comes from the fact that a compatible FT-ADMG would have a bidirected arrow $X_{t-\gamma} \dashleftarrow\dashrightarrow W_{t}$ which constitutes a backdoor path in the FT-ADMG and intuitively is potentially adversary to the application of the frontdoor.
> > > 2. Concerning the request for no cycles on X the *sufficiency* of the conditions in Def. 7: I wonder whether there are situations in which **there is** a cycle on $X$ in the SCG, yet the causal effect is identifiable by front-door. For example, consider the FT-ADMG
> > > $X_{t-1} \rightarrow W_t \rightarrow Y_t$, $X_{t-1} \dashleftarrow\dashrightarrow Y_t$ and $X_{t-1} \rightarrow X_t$. The last arrow introduces a cycle, but the front door still identifies $p(Y_t | do(X_{t-1}))$. Yet, the resulting summary graph would not satisfy conditions 4a and 4b of Def. 7. Correct?

---

> ### Author Response · Authors · 2025-02-13
> **Response to Reviewer k1g5 (second round)**
>
> * Regarding question 1:
> $X \leftrightarrow W$ is considered a backdoor path, meaning that Definition 7 does not cover cases where $X \leftrightarrow W$ is present. The reasoning aligns exactly with what you mentioned. However, we emphasize that even if backdoor paths are not allowed at the SCG level in Definition 7, they may still exist at the micro level, as explained in our previous response.
>
> * Regarding question 2:
> No, that is not correct (given our Definition of SCG), but thank you for asking. This point can indeed be very confusing, and clarifying it helps to highlight the complexity of causal effect identification using SCGs, thereby underscoring the contribution of the paper. Now, regarding the example: Yes, it is true that the front-door criterion is satisfied if we have access to the FT-ADMG (the micro-level graph). However, assuming we only have access to the compatible SCG, which consists of: $X\rightarrow W \rightarrow Y$ and $X \leftrightarrows Y$ with a self loop on X (In this formulation, we have replaced the bidirected edges with the latent variable $L_{t-1}$ to clarify our point.).
> In this case, knowing only the SCG—without any knowledge of the FT-ADMG—means that we must account for all possible FT-ADMGs that are compatible with the given SCG. One such FT-ADMG is: $X_t \rightarrow W_t \rightarrow Y_t$, $X_{t-1} \leftarrow L_{t-1} \rightarrow Y_t$, $X_{t-1} \leftarrow L_{t-1} \rightarrow X_t$, and $X_{t-1} \rightarrow X_t$.
> Notice that in this example, there exists a backdoor path: $X_{t-1} \leftarrow L_{t-1} \rightarrow X_t \rightarrow W_t$ that can only be blocked by $X_t$ but $X_t$ is a descendant of $X_{t-1}$ so we cannot adjust on it.
>
> Now, let’s consider another FT-ADMG that is compatible with the same SCG, where we have the following structure: $X_{t-1} \rightarrow W_t \rightarrow Y_t$, $X_{t} \rightarrow W_t \rightarrow Y_t$, $X_{t-1} \leftarrow L_{t-1} \rightarrow Y_t$, $X_{t} \leftarrow L_{t} \rightarrow Y_t$, and $X_{t-1} \rightarrow X_t$. In this case, it might seem that the front-door criterion is satisfied. However, since we are searching for a unique expression for identification and adhering to the expression given in the theorem, we would adjust for $X_t$. This poses an issue because $X_t$ acts as a collider between $X_{t-1}$ and $L_t$. Adjusting for a collider induces a correlation between $X_{t-1}$ and $L_t$, thereby activating the spurious path $X_{t-1} \dots L_t \rightarrow Y_t$, which introduces bias.
>
> However, we would like to point out that if we change the definition of SCG, specifically, allowing bidirected dashed edges between a node and itself, then your statement would be correct. If you think it is better to allow it we can do so and there would be an easy fix of Def 7 and Theorem 7:
> In Def7, we would add an item 4(c)  stating that Bidiirected self loop on X do not existe (so the causal effect is identifiable if the first three conditions are satisfied and if one of conditions 4(a), 4(b), 4(c) is satisfied.
> In Theorem 1: everything would remain the same but whenever Condition 4(c) is satisfied but Condition 4(a) and 4(b) are not satisfied then we remove from $\mathbb{B}^x$ the variables $(X_{t-\gamma+l})_{1\le l\le \gamma}$.
> Please let us know what you think.
>
> Thank you once again. Please let us know if you have any additional concerns, and we will be more than happy to address them.

---

> ### Comment · Reviewer_k1g5 · 2025-02-14
>
> Q1: don't you write in the **SCG notations** paragraph "a *backdoor path* between $X$ and $Y$ is a path between $X$ and $Y$ that starts either by $X \leftarrow$ or $X \leftrightarrows$"? In this sense, why would you consider $X \leftrightarrow W$ a backdoor path?
>
> ---
>
> Q2: thank you, the example perfectly clarified it. Now I also better understand the role of "uniquely" in definition 5, meaning that the causal effect is identifiable by theorem 1 if, whatever compatible FT-ADMG you choose, the adjustment formula of eq. (1) yields the same (correct) causal effect.

---

> > ### Author Response · Authors · 2025-02-17
> > **Response to Reviewer k1g5 (third round)**
> >
> > Oops, that was a typo. Any path that starts with an arrow pointing towards $X$ is a backdoor path. This means paths that begin with $X\leftarrow$ or $X\leftrightarrows$ or $X \leftrightarrow$. Thank you for pointing this out, we will correct it.

---

> > > ### Comment · Reviewer_k1g5 · 2025-02-20
> > >
> > > Thank you to the authors, all doubts are clarified.

---

### Review · Reviewer_QeaU · 2025-02-11

**Summary Of Contributions:**

The topic of the paper is identifiability of causal effects in graphical models that represent relationships among time series variables. Some literature has representing causal relationships among time series in a fine-grained way, with a separate set of variables for each time point, and other literature has instead used a coarse-grained summary graph representation, which abstracts away from some details of the dynamics. This work takes the latter approach, and investigates the conditions under which a causal effect may be identified in the presence of unmeasured confounding, focusing on the front-door strategy. The contribution is entirely theoretical: there is basically one main result which is a set of sufficient (but not necessary or complete) graphical conditions under which a causal effect is identifiable from the summary graph.

**Audience:**

No

**Broader Impact Concerns:**

None.

**Claims And Evidence:**

No

**Requested Changes:**

For the paper to be of interest to the TMLR readership, the result would need to be clearly distinguished from existing results and shown convincingly to somehow generalize or improve on existing identification theory. The other issues raised above would also need to be addressed.

**Strengths And Weaknesses:**

A fundamental weakness of the work is that a version of the front-door identifiability criterion has already been established in a very similar class of time series summary graphs by Eichler and Didelez (2010). They establish both back-door and front-door identification criteria. (The front-door result is their Theorem 4.12. A version of the result was stated without proof in an earlier conference paper, Eichler and Didelez (2007), which is cited by the authors of this manuscript, but this result is not mentioned and the 2010 paper where all details are spelled out is not cited at all.) I say that this result is "similar" but not "exactly the same" because Eichler and Didelez use a slightly different graphical set-up; they are also using mixed summary graphs (also allowing for cycles, only using dashed instead bidirected edges to represent unmeasured confounding) though the relationship between the Eichler and Didelez summary graph and the underlying "fine-grained" time series graphs is a bit different than the one assumed here. Eichler and Didelez assume no "contemporaneous" causal connections within a time-slice. The present authors allow contemporaneous connections, but assume these are acyclic. Besides this, I cannot really see any important difference between the setting here and the setting of Eichler and Didelez.* I could be wrong: maybe there is something that I am missing. But if that is correct then this manuscript does not seem to be adding much to the existing literature. The role of contemporaneous relationships is barely discussed in the manuscript. Perhaps if the authors made some theoretical argument as to what difference contemporaneous (acyclic?) connections make to the (non)-identifiability of causal effects across lags,** then at least the added contribution could be evaluated.

*I also think the Eichler & Didelez treatment is more rigorous, since they take into account some important statistical subtleties about the underlying time series processes that the authors here gloss over or do not address (see below).

**Obviously, if the effect of interest is purely contemporaneous, i.e., of X_t on Y_t, then there is nothing new being added, because the front-door criterion in that case is just exactly the same as Pearl's. It is only lagged effects that are of interest here: X_{t-h} on Y_t for some h.

Other weaknesses:

- It seems that the only real difference between the result proposed here and existing results is the last condition in the modified front-door criterion, which requires Cycles(X, G^s) = \emptyset. This is not really discussed in any detail or explained, and the consequences of this are not very clear to me. First, it is unclear if the authors include "self-cycles" in this definition. Many of the example graphs have X_{t-1}-->X_t edges and thus X-->X cycle in the summary graph. In the caption to Figure 1(g), the authors say their criterion doesn't hold but the only cycle is a self-cycle, so I guess X-->X means Cycles(X, G^s) = X and that is what precludes identifiability here. But then the criterion basically does not apply to any graph where the exposure variable causes itself over time, which is arguably most realistic dynamical systems. It is also well-known in this literature that X-->X and X<-->X are not distinguishable from each other empirically. (If there are such self-cycles, then the criterion only works for contemporaneous not lagged effects, which is just Pearl's criterion as mentioned at ** above. The examples in Figure 3 are all graphs where the effect of interest is X_t on Y_t and so it is just Pearl's front-door.) If I am misunderstanding something about the criterion and self-cycles are allowed, please correct me. But if I am correct in my understanding, then the result as stated is far too restrictive to be useful. And notably the Eichler and Didelez result does not make this restriction, or at least not in the same way.

- The definition of the underlying dynamical model leaves out many important details. Definition 1 does not make clear if the (full) process is of finite Markov order, or if infinite order processes are allowed. Even with finite Markov order in the full process, there may be no finite order process over the observed marginal because of "infinite lag" association induced by confounding. Assumption 2 says that the functional relationships to not depend on time, but this is not sufficient to imply stationarity of the joint distribution over all finite collections of random variables. There are additional conditions required such that the induced distributions behave well. For example, see conditions (T1-T3) and (I1-I4) in Eichler and Didelez (2010).

- Pg. 1: Describing "experimentation" as simply "free from bias" is not accurate. There is a huge literature on possible sources of bias in randomized experiments.
- Pg 2: Pearl's front-door criterion does not assume a DAG over the observed variables, it assumes an ADMG (or what he calls a semi-markovian model).
- Pg 5: the discussion of learning time series graphs from observational data is much too quick and dismissive. What does "not satisfactory" mean? What are the "non-testable strong assumptions"? I agree that causal discovery from time series is challenging, but it is not obvious that the challenges are due primarily to some "strong assumptions" or other reasons. The brief sentences here and footnote do not fairly reflect the large literature on causal discovery from time series.
- First paragraph on pg 7 is not written very clearly. What does "there is a backdoor path between Xt−1 and Wt passing by Xt which should not be blocked by Xt" mean? ("should not be"?)
- I only skimmed the technical Lemmas briefly and they seem mostly sound, but some parts of the proofs are written in a confusing way that makes them hard to follow. For example, in the proof for Property 1: "Since the path does not pass through another instance of X, the compatible of the path considered in Gs is the following path..." The proofs could be cleaned up.
- There are many other points in the text with typos or unclear writing, which would benefit from some careful editing.
- There is other existing literature on causal identification theory for dynamical systems that should be cited. For example, in addition to the work of Eichler and Didelez, there are papers by (e.g.) Niels R Hansen and Søren W Mogensen et al on causal representations of stochastic processes, there are papers by Joris Mooij, Jonas Peters, et al on identification in cyclic graphs, related papers by Jakob Runge et al on identification using certain other time series representations, etc.

Refs:
Eichler, M., & Didelez, V. (2010). On Granger causality and the effect of interventions in time series. Lifetime data analysis, 16, 3-32.

---

> ### Author Response · Authors · 2025-02-13
> **Response to Reviewer QeaU (1/4)**
>
> We thank the Reviewer for all the remarks and comments.
> We start by addressing the main concern of the reviewer regarding the contribution of the paper. Then,  we address all other remarks and questions one by one.
>
> ## Addressing the comments regarding contributions of the paper:
>
> First, we would like to highlight that the identification of causal effects in the setting considered by Eichler and Didelez is much closer to the classical framework introduced by Pearl than to our setting. The key distinction lies in the fact that we account for contemporaneous relations (which makes cycles challenging), whereas Eichler and Didelez do not. By neglecting contemporaneous relations, it becomes straightforward to transform a summary causal graph into a dense acyclic graph containing the true graph, making the work of Pearl applicable. Since the summary graph and its acyclic transformation are equivalent under the assumption of no contemporaneous relations, cycles do not pose a major difficulty in that setting. For example, in their framework, if $X$ causes $Y$ and $Y$ causes $X$ in the summary graph (without any confounding), then we can construct an equivalent dense acyclic graph (supposing the max lag =1) where  $X_{t_1}$ causes $X_{t}$, $Y_{t_1}$ causes $Y_{t}$, $X_{t_1}$ causes $Y_{t}$ and $Y_{t-1}$ causes $X_{t}$ (this is not necessarily the true causal graph, ie, the true causal graph can either be this graph or a subgraph of this graph). Since this transformed graph is acyclic, it becomes evident why the back-door criterion can be directly applied, i.e. we can estimate the effect of $X_{t-1}$ on $Y_t$ by adjusting on the past of $X_{t-1}$. Similarly, we can construct examples illustrating why it is obvious that the front-door criterion is satisfied in that setting.
> However, in our framework, where contemporaneous relations exist, these transformations are no longer valid, making the identification problem fundamentally more complex (and cycles becomes as much a challenge as hidden confounding). To see why, let’s take the same summary graph where $X$ causes $Y$ and $Y$ causes $X$ but without assuming no contemporaneous relations. In this case, there is not a unique acyclic graph that preserves all information in the summer graph. For instance, we can imagine an acyclic graph G1 (compatible with the summary graph) where  $X_{t_1}$ causes $X_{t}$, $Y_{t_1}$ causes $Y_{t}$, $X_{t-1}$ causes $Y_{t}$, $Y_{t-1}$ causes $X_{t}$, $Y_{t-1}$ causes $X_{t-1}$  and $Y_{t}$ causes $X_{t}$. At the same time we can imagine an acyclic graph G2  (compatible with the same summary graph) where  $X_{t_1}$ causes $X_{t}$, $Y_{t_1}$ causes $Y_{t}$,, $X_{t-1}$ causes $Y_{t}$, $Y_{t-1}$ causes $X_{t}$, $Y_{t-1}$ causes $X_{t-1}$  and $Y_{t}$ causes $X_{t}$. As you can see G1 and G2 are cannot be considered as subgraph of the same acyclic graph. For the same reason, in G1 the causal effect  $X_{t-1}$ on $Y_t$ can be estimated by adjusting on the past of $X_{t-1}$ (i.e., $X_{t-2}, Y_{t-2}$). But in G2, to estimate the causal effect we need to also adjust on $Y_{t-1}$ since $Y_{t-1}$ is a confounder between $X_{t-1}$ and $Y_{t}$. However, we should not adjust on $Y_{t-1}$ in G2 because in this case $Y_{t-1}$  is an intermediate cause between $X_{t-1}$ and $Y_{t}$. As you can see, in this case, there is no unique solution to identify the causal effect. Therefore, it is not possible to apply the back-door criterion in this setting (even without hidden confounding!).

---

> > ### Author Response · Authors · 2025-02-13
> > **Response to Reviewer QeaU (2/4)**
> >
> > Second, we would like to emphasize that, similar to Eichler and Didelez, we assume that the true causal graph is acyclic. However, also like Eichler and Didelez, we allow for cycles in the summary graph, which is the only available representation since the true causal graph is unknown. Thus, while we permit cycles at the macro level, our approach differs from theirs in that we account for instantaneous relations. Note that even when the causal effect of interest is lagged, our result is different than Eichler and Didelez, since even if the causal effect of interest is lagged, in the true causal graph there might be instantaneous relations and so there are many difficulties like the one presented above.
> > We do not believe that our work can simply be reduced to Pearl's framework or to Eichler and Didelez's approach or both comboned. Both lagged effects and instantaneous effects are important and pose significant challenges in our setting:
> > 1) When the causal effect of interest is lagged, our results differ from those of Eichler and Didelez.  For example, by directly applying their front-door (introduced when there are no instantaneous relations) on the summary graph in Figure 1 (g) (assuming instantaneous relations), you might think that the causal effect of $X_{t-1} on $Y_t$ is identifiable by front-door but it is not (this can be demonstrated using the same logic as in Section 5).
> > 2) When the causal effect of interest is lagged, our results differ from those of Pearl too. For instance, in Figure, 2(a) even thought it seems that we can use the classical front-door (because of acyclicity) to identify the causal effect, this is not entirely true: it is true that the conditions of the classical font-door are satisfied, it is true the causal effect is indeed identifiable (given Def 7), but the identification formula that should be used cannot be given by the formula given in the by Pearl in the context of the front-door and DAGs (because hidden confounding in the summary graph can imply many other hidden confoundings in the true causal graph). The formula that should be used is the one given in our Theorem. For instance, from the classical front-door definition and the classical front-door formula, you would not be able to guess that you need to include  $X_{t}$ in formula to identify the effect of $X_{t-1}$ on $Y_t$;
> > 3) When the causal effect of interest is instantaneous, obviously we are different from our results differ from Eichler and Didelez;
> > 4) When the causal effect of interest is instantaneous, our results also differ from Pearl’s framework because we can have cycles (cycles on $X$ can imply many backdoor paths between $X_{t-1}$ and $Y_t$, more details on this was given in our response to Reviewer k1g5).
> > In summary, we do not think that your work can be reduced to the work of Pearl or to the work of Eichler and Didelez or both combined,  even when graphically they seem to be equivalent. To appreciate the difference one need to look at the graphical conditions as well as at the do-free expression in the theorem.
> >
> > Third, yes, our results can sometimes be quite restrictive, particularly when analyzing lagged effects, as in that case Def7 requires the absence of cycles on  $X$. However, this is a necessary constraint to apply the front-door in our setting (using summary graph and assuming instantaneous relations). In cases when there are cycles on $X$ and you have instantaneous relations, it is not possible to apply nor to extend the classical front-door criterion correctly to identify the effect of $X_{t-1}$ on $Y_t$. Even, the work of Eichler and Didelez is not applicable in this case, further highlighting the limitations of existing approaches in our setting.
> >
> > In conclusion, the contribution of this paper is clearly outlined in the introduction: «  in the setting where a fully specified temporal causal graph is not available, this paper addresses the challenge of identifying total effects of one variable within a cluster on another variable in a different cluster when having access to an SCG while allowing instantaneous relations, cycles, and latent confounding. » If you are aware of any paper that provides similar results for the specific problem we are addressing, please let us know—we would be more than happy to explore the differences. As previously discussed, neither Pearl’s work, nor that of Eichler and Didelez, nor a combination of both, directly addresses this problem. Likewise, if you know of any paper that presents more general results from which our findings could be derived, we would also be very interested in examining the distinctions. Again, as discussed earlier, this is not the case for the works of Pearl or Eichler and Didelez, either separately or together.

---

> > > ### Author Response · Authors · 2025-02-13
> > > **Response to Reviewer QeaU (3/4)**
> > >
> > > ## Addressing various remarks and questions not related to the one addressed above:
> > >
> > > We start by restating the question/comments and then we give our response.
> > >
> > > * The definition of the underlying dynamical model leaves out many important details. Definition 1 does not make clear if the (full) process is of finite Markov order, or if infinite order processes are allowed. Even with finite Markov order in the full process, there may be no finite order process over the observed marginal because of "infinite lag" association induced by confounding. Assumption 2 says that the functional relationships to not depend on time, but this is not sufficient to imply stationarity of the joint distribution over all finite collections of random variables. There are additional conditions required such that the induced distributions behave well. For example, see conditions (T1-T3) and (I1-I4) in Eichler and Didelez (2010).
> > >
> > > **Response**
> > > We acknowledge that we were not entirely formal regarding certain details of the model, and we apologize for that. However, part of this was intentional. First, to clarify, we assume a finite Markov order, and we will make this explicit in the definition. Beyond this assumption, the only additional conditions we impose are those inferred from the graph, the positivity assumption on distributions (as stated in Definitions 3 and 5), and a finite $\gamma_{max}$ that is smaller than the length of the series. Other than these, no further assumptions are required. Our goal was to develop a general model that remains a specific case of Pearl’s SCM. If we only have one observation per node (i.e., in the case of a single time series realization), then additional assumptions—namely one you proposed—are indeed necessary for causal effect estimation. However, if we can collect multiple observations for each node, then no extra assumptions are required. We will clarify all of this in the revised version and cite these assumptions in case of having one observation of each node. Thank you for pointing it out!
> > >
> > > * Pg. 1: Describing "experimentation" as simply "free from bias" is not accurate. There is a huge literature on possible sources of bias in randomized experiments.
> > >
> > > **Response**: We agree with you. What we meant to say here is « free from confounding bias » in an ideal scenario. We will rectify this.
> > >
> > >
> > > * Pg 2: Pearl's front-door criterion does not assume a DAG over the observed variables, it assumes an ADMG (or what he calls a semi-markovian model).
> > >
> > > **Response**: What we are trying to point out here is that Pearl’s entire approach is built on DAGs (the considering both endogenous and exogeneous variables are nodes, i.e., even when it is a semi-Markovian model). Even when working with  ADMGs (which are latent projection of DAGs, that contain only endogenous variables), these can always be conceptualized in terms of underlying DAG structures. In some papers Pearl uses the notions of ADMGs and in other (for example in his 2009 book Figure 3.5), he uses DAGs in an equivalent manner (where he highlights what are the hidden variables in the DAG). We will clarify this.
> > >
> > >
> > > * Pg 5: the discussion of learning time series graphs from observational data is much too quick and dismissive. What does "not satisfactory" mean? What are the "non-testable strong assumptions"? I agree that causal discovery from time series is challenging, but it is not obvious that the challenges are due primarily to some "strong assumptions" or other reasons. The brief sentences here and footnote do not fairly reflect the large literature on causal discovery from time series.
> > >
> > > **Response**: First, here when talking about the strong assumptions we meant to talk about causal discovery in general. Not just for time series. In the domain of time series, this motivates working with summary graphs because they are easier to construct manually compared to the true causal graphs.
> > > We will cite more papers about causal discovery (including what the reviewer proposed in addition to surveys  but  it is hard to find papers saying that causal discovery is very challenging due to the strong assumptions (because usually accepted papers focus on positive results). Nevertheless, the inherent complexities and challenges of causal discovery are well-recognized in the field (that is also one of the reasons why it is such an interesting field).
> > > We will also rephrase the sentence as follows:  « Furthermore, the assumptions that causal discovery methods rely on to identify the true FT-ADMG from real observational data are not always satisfied in practical applications. » Does this seem more acceptable to you?

---

> > > > ### Author Response · Authors · 2025-02-13
> > > > **Response to Reviewer QeaU (4/4)**
> > > >
> > > > * First paragraph on pg 7 is not written very clearly. What does "there is a backdoor path between Xt−1 and Wt passing by Xt which should not be blocked by Xt" mean? ("should not be"?)
> > > >
> > > > **Response**:
> > > > What we meant to say is that the path can only be blocked by $X_t$ but adjusting on  $X_t$ in this case would bias the causal effect since  $X_t$  is an intermediate cause between $X_{t-1}$ and $Y_t$. We will clarify this point.
> > > >
> > > >
> > > > * I only skimmed the technical Lemmas briefly and they seem mostly sound, but some parts of the proofs are written in a confusing way that makes them hard to follow. For example, in the proof for Property 1: "Since the path does not pass through another instance of X, the compatible of the path considered in Gs is the following path..." The proofs could be cleaned up.
> > > >
> > > > **Response**:
> > > > We agree. We will clean it and make it clearer.
> > > >
> > > >
> > > > * There is other existing literature on causal identification theory for dynamical systems that should be cited. For example, in addition to the work of Eichler and Didelez, there are papers by (e.g.) Niels R Hansen and Søren W Mogensen et al on causal representations of stochastic processes, there are papers by Joris Mooij, Jonas Peters, et al on identification in cyclic graphs, related papers by Jakob Runge et al on identification using certain other time series representations, etc.
> > > >
> > > > **Response**: We will gladly cite all the papers you mentioned. However, we would like to clarify our position on their relevance to our work. Yes, the journal version of Eichler and Didelez should definitely be cited and discussed, as their work is highly related to ours (but again they do not assume instantaneous relations which is the biggest challenge in our setting). Similarly, the papers by Jonas Mooij on causal identification using cyclic graphs are also relevant, even though they do not consider the same setting and their results are not directly applicable here. However, as far as we know, there are no papers by Niels R. Hansen, Søren W. Mogensen, or Jakob Runge that are directly related to the specific problem we are addressing. That said, we will cite their work when discussing causal discovery methods. If we are mistaken, and you believe that these authors have contributed to the identification of causal effects using some form of cluster graph, please let us know. In that case, we will be happy to examine their contributions further and discuss them accordingly (but again we precise that in any case we will cite them when discussing causal discovery, which is not center of our work).
> > > >
> > > > We hope our responses adequately address all concerns, and we look forward to any further feedback.

---

> > > > ### Comment · Reviewer_QeaU · 2025-03-05
> > > > **Reviewer response to authors**
> > > >
> > > > The authors argue here that the key difference between their setting (allowing contemporaneous relations) and the one in Eichler & Didelez makes a difference for what is identified and how. This is not discussed at all in the paper (again, Eichler and Didelez's result is not even cited) and certainly not described with a clear convincing example for the reader. So, the essential case that I would request in order to recommend this work for publication is the following: step through a simple example (say the graph in Fig 1(g) or similar) where front-door identification of a lagged effect "succeeds" using the Eichler & Didelez criterion/assumptions and where front-door "fails" allowing for contemporaneous relationships. This would involve a thorough descriptive comparison of the two criteria and this should be central to the presentation of the paper. [I don't think the reverse is true, that there will be graphs where something is identified by your criterion/assumptions and not identified by theirs, unless I'm missing something, but if there were such a case it would be important to also highlight and go through this.] The example in the first comment above with X and Y in a cycle is not quite what is needed here, since this concerns back-door rather than front-door. It is hard for me to recommend the paper without seeing such a revision.
> > > >
> > > > I don't follow point 4 above: it would be good to also walk through an example (different from above) for purely instantaneous effects where the criterion here differs from what you get from Pearl's criterion.
> > > >
> > > > In the reply here the authors describe the "no self-cycles" condition in their theorem as "necessary" but it should be carefully distinguished what is necessary from sufficient. The whole front-door criterion describes a *sufficient* condition for a quantity to be identified. Perhaps the authors mean that "without a prohibition on self-cycles on X, the front-door formula we state cannot apply," but it is entirely possible that even in the presence of self-cycles, effects may be identified by other means (using the full power of the ID algorithm). In any case, if it is true that with self-cycles the front-door formula described here will never apply, then this severely limits the applicability of this identification strategy, since in real dynamical systems usually variables states depend on their past states. A simple example explaining why self-cycles preclude front-door-identification should be added.
> > > >
> > > > The setting where one can collect multiple observations per node is very different from the setting with one observation per node (standardly called multiple time series) and these should be distinguished/clarified. My understanding is that the paper was about the latter setting (one observation per node) which is why the formal conditions on the stationarity of the stochastic processes are necessary. If the results are meant to apply in *both* settings then I am quite confused.
> > > >
> > > > Re: causal discovery sentence. My objection is to making statements about "the assumptions" and their non-satisfaction without any reference to what "the assumptions" are, since these are different for different methods and whether some assumptions are satisfied depends on the context -- any such statement is vague. Instead, I would suggest citing the relevant causal discovery from time series literature and simply stating that causal discovery is challenging. The theoretical work here should be valuable independent of what one thinks about the feasibility of causal discovery. (Note: there are not generally methods for causal discovery of ADMGs for reasons of non-identifiability - in settings with latent confounding, the target is usually something like a PAG.)
> > > >
> > > > Re: additional references, here is one work by Runge that is not only about causal discovery, but also about parameter identification (in a setting with contemporaneous relationships, an SVAR): Reiter, Nicolas-Domenic, Jonas Wahl, Andreas Gerhardus, and Jakob Runge. "Causal inference on process graphs, part II: Causal structure and effect identification." arXiv preprint arXiv:2406.17422 (2024).

---

> > > > > ### Author Response · Authors · 2025-03-06
> > > > > **Response to Reviewer QeaU (1/2) (Second round)**
> > > > >
> > > > > Thank you for your taking the time to reply to our comments we think that our discussion helped improve a lot the clarity of the paper and at the same time the contribution. We will upload a new version of the paper where all modifications are in red (these modifications address comments of all reviewers).
> > > > >
> > > > > In section 3, we clarified (using intuition and using examples) why the standard front-door criterion and its extension to time series (when there are no instantaneous relations in the system) cannot be applied in our context. Let us summarize these examples here.
> > > > > * The first example is when there is a self loop on $X$ and when there is also a self bidirected dashed edge as in Figure 1 (i) (in the new version of the paper). Suppose we are interested in the effect of $X_{t-1}$ on $Y_t$. By looking at the SCG, it seems that $W$ satisfies the standard font-door criterion and its extension in Eichler and Didelez. Which means intuitively, $W_{t-1}$ and W_{t} (we will call this the font door set) should be used in the front-door formula as the the set of variables that fully mediates the effect $X_{t-1}$ on $Y_t$.
> > > > > However, in this case, we can imagine an FT-ADMG compatible with the SCG where we have the backdoor path $X_{t-1} <-> X_t \rightarrow W_t$. Obviously, since we are using $W_t$ in the front-door set, we need to block this path. The only way to block it is by adjusting on $X_t$. However, given the SCG, we can also imagine in the same compatible FT-ADMG, that we have the directed path $X_{t-1}\rightarrow $X_t \rightarrow W_t$. This means that we should not adjust on $X_t$. Therefore we conclude that the $W_t$ should be in the font door set, therefore in this case we cannot use the font-door formula (W_{t-1} alone cannot be used since it does not fully mediate the effect).
> > > > > * The second example is when there is a cycle on $X$ which is bigger than a self loop. For example, like the one given in Figure 3 (b), where there is a cycle that includes $X$ and $U$. For simplicity we can disregard bidirected dashed loop on $X$ (remove $X <-> X$ from the SCG). In this case, as before, by looking at the SCG, it seems that $W$ satisfies the standard font-door criterion and its extension in Eichler and Didelez. Which means intuitively, $W_{t-1}$ and W_{t} (we will call this the font door set) should be used in the front-door formula as the the set of variables that fully mediates the effect $X_{t-1}$ on $Y_t$. However here we can imagine two different FT-ADMGs compatible with the SCG such that in the first we have the backdoor path $X_{t-1}\leftarrow U_{t-1} \rightarrow X_t$ (therefore we should adjust either on $U_{t-1}$ or $X_t$) in the first FT-ADMG but where in the other FT-ADMG this path is directed path $X_{t-1}\rightarrow U_{t-1} \rightarrow X_t$ (therefore we should NOT adjust on $U_{t-1}$ or $X_t$). Since we do not know which one is the true FT-ADMG, we conclude that we cannot use $W_t$ in the front-door set. The standard front-door criterion and its extension in  Eichler and Didelez are not equipped to handle such cases and so using them in our context one might think that the total effect is can be identified by using $W_{t-1}$ and W_{t} in the front-door set.
> > > > >
> > > > > Regarding instantaneous relations. Let us be clear the big differences with Pearl’s front-door criterion is  not in this case. In this case, there is a difference with Pearl’s front-door criterion but it is very subtle. First we clearly state that $W$ satisfies the our criterion for ($X_t, Y_t$) iff it satisfies Pearl’s front-door criterion. The difference is not in whether the causal effect is identifiable or not. But rather in how to identify it. Let’s consider the SCG in Figure 2 (b) satisfying both criterions and consider the total effect of $X_t$ on $Y_t$. In this case, there are not backdoor path between $X$ and $W$ in the SCG, However, it is possible to imagine an FT-ADMG where we have the backdoor path $X_t\leftarrow U_t \leftarrow … \leftarrow U_{t-1000} \rightarrow X_{t-1000} \rightarrow W_{t-1000} \rightarrow … \rightarrow W_{t}$. This path should be blocked. To blocked this path and all similar paths you can either adjust on all variables between X_{t-2000}  and X_{t-1} (You cannot only adjust on X_{t-1000}, since there might be other similar paths and we do not have the true FT-ADMG), assuming t-2000 is the oldest timepoints in the time series (this is not pratical) or you need simply adjust on $U_{t}$ $U_{t-1}$, … $U_{t-\gamma_{max}}. Pearl’s front-door criterion  and the  theorem associated to it cannot be used to figure this out.

---

> ### Author Response · Authors · 2025-03-06
> **Response to Reviewer QeaU (2/2) (Second round)**
>
> You are absolutely right in your remark regarding our use of the term "necessary." We apologize for the confusion. As you correctly pointed out, what we intended to convey was that the front-door formula we presented does not apply in the given context. However, as suggested by Reviewer k1g5, we have incorporated a new condition into our criterion that allows self-loops, provided that there are no bidirected dashed edges between $X$ and its ancestors. In this case, the formula is slightly modified: we simply do not adjust on the future of $X_{t-\gamma}$. As a result, our revised paper now includes cases where self-loops are permitted in some cases, even in the absence of instantaneous relations.
>
> Thank you so much for the reference. It is very relevant! We included it.
>
> Once again, we sincerely appreciate your time and effort in helping us improve the quality of our paper. We hope that our response has addressed all your concerns. However, if any points remain unclear or if you have any further questions, we would be happy to continue this discussion.

---

### Review · Reviewer_g9tu · 2025-02-26

**Summary Of Contributions:**

This paper provides a front-door criterion specifically tailored for Summary Causal Graphs (SCGs). SCGs serve as a useful abstraction when there exists causal dependencies between temporal variables. In SCG, those temporal variables are considred as a single vertex—thereby abstracting away detailed temporal lags. The paper demonstrates that the standard Pearl's front-door criterion, which is applicable to fully specified causal graphs (such as FT-ADMGs), is not directly applicable when working with SCGs. To address this limitation, the authors develop an SCG-specified front-door criterion that includes additional conditions (e.g., no cycles in treatment-temporal variables or no time lag in the treatment).

**Audience:**

No

**Broader Impact Concerns:**

This paper doesn't have any ethical implications.

**Claims And Evidence:**

No

**Requested Changes:**

1. **Working Example:**
   Add a fully worked-through example that starts with an FT-ADMG and shows how it is projected into an SCG. Use this example to illustrate a scenario where the standard front-door criterion fails—because latent backdoor paths or cycles exist—but the SCG front-door criterion (with its additional no-cycle constraint on \(X\)) succeeds. This example should be central to the paper and help readers see the practical impact of the theoretical contribution.

2. **Clarify Definitions and Notation:**
   - Revise Definition 4 to clearly explain that \(S := \{Y \mid \forall Y_t \in V\}\) means that \(S\) is a set of clustered variables, where each element \(Y\) represents a collection of temporal observations \(\{Y_t\}\).
   - Provide an explicit definition of “macro vertices” in Definition 6 and explain how they differ from the micro (time-indexed) vertices.

3. **Strengthen the Motivation by adding Real-world Examples:**
   Include a real-world inspired case study that shows why working with SCGs is useful—especially when the detailed temporal information is hard to specify. Clearly explain the limitations of applying Pearl's standard front-door criterion in dynamic settings and why the added constraint in the SCG front-door criterion is necessary.

4. **Improve Accessibility:**
   Update the text by not using colors like purple or brown so that color choices are easier to distinguish. Consider using colors that are friendly to color-blind readers or adding patterns/labels to ensure the figures are accessible.

5. **Address Novelty Concerns:**
   Acknowledge that the additional constraint (e.g., requiring \(Cycle(X, G_s)=\emptyset\) or \(\gamma=0\)) might seem incremental. Explain in more detail how this condition is essential for identifying the total effect in SCGs and why it matters in practice. You may highlight the importance or significant through designing examples.

**Strengths And Weaknesses:**

**Strengths:**
- The paper offers a rich literature review that effectively summarizes the historical development of theories related to causal inference, particularly in the context of partially specified graphs.
- It provides a solid proof sketch for its main results, and all technical derivations appear sound.
- The work is technically self-contained, making it accessible to readers who may not be deeply familiar with every aspect of the underlying literature.

**Weaknesses:**
- The paper does not provide any explicit working example that shows an SCG derived from an FT-ADMG in which the SCG front-door criterion is satisfied. This absence makes it difficult to assess the practical relevance and applicability of the theoretical results.
- The definitions and notations are sometimes unclear. For example, in Definition 4 the set $S := \{{Y \mid \forall Y_t \in V \}}$ is ambiguous, leaving it uncertain whether $Y$ represents a cluster of temporal observations. Similarly, the notion of "macro vertices" in Definition 6 is not explicitly defined.
- The motivational aspects of the paper are weak. Although the paper reviews the literature extensively, it lacks a real-world or empirically driven working example that demonstrates the utility of the proposed approach.
- The presentation could be improved for accessibility; for instance, the color choices in the figures (e.g., brown, purple, gray) might not be easily distinguishable for color-blind readers.
- To my understanding, the SCG front-door criterion essentially adds the constraint that there must be no cycle involving \(X\) (or equivalently, \(Cycle(X, G_s) = \emptyset\))—a condition needed to avoid back-door paths from \(X_t\) to \(X_{t'}\). If this is indeed the case, the contribution appears somewhat incremental rather than truly novel. I recommend that the paper be reorganized to more clearly highlight a concrete example where the standard front-door criterion fails, but the SCG-specified front-door criterion succeeds. A fully worked-through example that penetrates the entire paper would greatly enhance both the clarity and impact of the contribution.

---

> ### Author Response · Authors · 2025-03-04
> **Response to Reviewer g9tu (1/2)**
>
> We thank the Reviewer for constructive and valuable comments. Detailed responses to all the points raised by the Reviewer are reported below.
>
> ## Addressing the comments in Weaknesses and Questions
>
> Here, we address the comments and questions raised in the Weaknesses section and in the requested changes section. We begin by restating each comment before providing our response.
>
> * Weakness 1 and requested Change 1: The paper does not provide any explicit working example that shows an SCG derived from an FT-ADMG in which the SCG front-door criterion is satisfied. This absence makes it difficult to assess the practical relevance and applicability of the theoretical results.
>
> **Response**:
> The first figure was intended to illustrate a working example of an SCG where the SCG-front-door criterion holds when $\gamma = 0$. In this SCG, the standard front-door criterion seem to be satisfied because all its graphical conditions are satisfied; however, its associated do-free formula cannot be applied because there exist backdoor paths between $X_{t-\gamma}$ and $W_{t-\gamma}$ that are not detectable solely by examining the SCG. We acknowledge that this figure was not utilized effectively, and we will address this issue. Instead of presenting a single SCG, we will revise the first figure to include three distinct SCGs, each highlighting a specific challenge that prevents the application of the standard front-door criterion. Additionally, for each SCG, we will provide two compatible FT-ADMGs to better illustrate the underlying difficulties.
>
> * Weakness 2 and requested Change 2: The definitions and notations are sometimes unclear. For example, in Definition 4 the set 𝑆 is ambiguous, leaving it uncertain whether Y represents a cluster of temporal observations. Similarly, the notion of "macro vertices" in Definition 6 is not explicitly defined.
>
> **Response**: We will revise the definition to clarify this point. Informally, each vertex in an SCG represents a cluster of all temporal variables belonging to a single time series, meaning that a vertex corresponds to an entire time series. A macro vertex is simply a vertex in an SCG (as stated on page 6). Since we will clarify the definition of a vertex in an SCG, this will also inherently clarify the meaning of a macro vertex.
>
>
> * Weakness 3 and requested Change 3: The motivational aspects of the paper are weak. Although the paper reviews the literature extensively, it lacks a real-world or empirically driven working example that demonstrates the utility of the proposed approach.
>
> **Response**:
> We will connect one the SCGs in the working example to a simplified real-world scenario where $X$ represents sedation levels, $Y$ represents arterial pressure, and $W$ corresponds to heart rate variability. However, this example will serve purely as an conceptual example rather than a real application with empirical data—similar to Pearl’s 1995 example on smoking and cancer, which was obviously a conceptual case. It took nearly 25 years before the front-door criterion found a concrete application in healthcare and epidemiology. Our example will be situated in the domain of critical care, where we assume an interest in assessing the effect of administering a treatment of sedation levels on blood pressure for a given patient in critical care. We consider a setting where daily patient monitoring records sedation levels, arterial pressure, and heart rate variability over time. In the SCG, we assume the following causal relationships: The time series sedation levels affect heart rate variability. Heart rate variability influences arterial pressure. The effect of sedation levels on arterial pressure is fully mediated by heart rate variability (i.e., no direct causal effect of sedation levels on arterial pressure). We assume the presence of a latent confounder (physiological factor) between sedation needs and blood pressure regulation, and latent confounding between different time points of $X$ and latent confounding between different timepoints of $Y$. No additional latent confounders exist. In this scenario, applying the standard front-door criterion might incorrectly suggest that the classical do-free formula associated with the front-door criterion can be used. However, using our SCG-front-door criterion and theorem, we detect the correct do-free formula.
>
>
>
> * Weakness 4 and requested Change 4: The presentation could be improved for accessibility; for instance, the color choices in the figures (e.g., brown, purple, gray) might not be easily distinguishable for color-blind readers.
>
> **Response**: We have updated the color palette to one that is accessible to color-blind readers.

---

> > ### Author Response · Authors · 2025-03-04
> > **Response to Reviewer g9tu (2/2)**
> >
> > * Weakness 5 and requested Change 5: To my understanding, the SCG front-door criterion essentially adds the constraint that there must be no cycle involving (X) (or equivalently, (Cycle(X, G_s) = \emptyset))—a condition needed to avoid back-door paths from (X_t) to (X_{t'}). If this is indeed the case, the contribution appears somewhat incremental rather than truly novel. I recommend that the paper be reorganized to more clearly highlight a concrete example where the standard front-door criterion fails, but the SCG-specified front-door criterion succeeds. A fully worked-through example that penetrates the entire paper would greatly enhance both the clarity and impact of the contribution.
> >
> > **Response**:
> > The SCG from-door adds the constraint that there must be no cycle involving X OR \gamma=1. In addition, we will add the following third constraint, as Reviewer suggested: OR no bidirected dashed arrow on $X$.
> > If either one of these constraints is satisfied in addition to the constraint of the standard front-door criterion then the causal effect is identifiable. However, It is important to note that even if one of these constraints is satisfied and the causal effect is identifiable, it cannot be identified using the standard do-free formula associated with the front-door criterion (as introduced in Pearl, 1995), because there will still be some back-door paths between $X_{t-\gamma}$ and the temporal variables compatible with $W$ that cannot be detected by only looking at the SCG. For example, we can imagine an FT-ADMG compatible with the SCG in Figure 2(b) where there exists the backdoor path $X_{t-1}\lefttarrow U_{t-1}\rightarrow X_{t}\rightarrow W_{t}$. Notice that this backdoor path is not visible by only looking at the SCG in Figure 2(b). As said before, we will make better use of working example (which will be also modified).
> >
> > We hope our responses adequately address all concerns, and we look forward to any further feedback.

---

### Decision · Action_Editor_2ZxQ · 2025-04-22

**Recommendation:** Accept with minor revision

**Comment:**

The reviewers disagreed on the assessment. I suggest that the paper be accepted with minor revisions, on the conditions:
* an additional paragraph is added highlighting the differences with Eichler & Didelez (2010), including at least one example where contemporaneous effects yield different results
* A general improvement in the clarity, especially around the point of one vs multiple observations per time series.
* Addition of a clear practical example showing an SCG derived from an FT-ADMG where the standard front-door fails but the SCG version succeeds (weakness 1 and change 1 from g9tu -- the reviewer stated in the current revision this was not yet implemented)

**Audience:**

This paper is relevant for the causal inference and dynamical systems communities, hence a good fit for TMLR

**Claims And Evidence:**

This paper introduces a methodology to estimate causal effects in partially specified graphs. They focus on summary causal graphs, common in dynamical systems. Interestingly, these also allow cycles and confounding, for which this paper present graphical conditions for identifiability.

---

> ### Author Response · Authors · 2025-04-25
>
> Thank you very much for accepting the paper. We just have a few clarifications regarding the requested minor revisions.
>
> Could you please confirm whether the comments for revision are based solely on the initial submission, or if they also take into account the revised version (where modified paragraphs are in red) we submitted in response to the reviewers’ comments?
>
> We ask because we believe the concerns raised have already been addressed in the revised manuscript:
>
> Point 1: This was addressed in Section 3 (pages 6 and 7, paragraphs marked in red). We included examples where Pearl’s front-door criterion and Eichler and Didelez’s front-door criterion fail for non-instantaneous relations. We also gave an example where Pearl’s front-door fails for instantaneous relations. While we did not explicitly mention Eichler and Didelez’s approach in the case of instantaneous relations, we indicated in Section 1 (page 2, paragraph in red) that their method does not apply to such settings (they explicitly only considered non-instantaneous relations).
>
> Point 2: This was considered in Section 2 (page 3, last paragraph in red).
>
> Point 3: Addressed in Section 4 (pages 14–15), where we added a realistic example involving the effect of sedation levels on blood pressure, with the summary causal graph shown in Figure 1c. In this scenario, the SCG-front-door criterion successfully identifies the causal effect, while the standard front-door criterion does not.
>
> Could you please let me know if these revisions are sufficient, or if additional modifications are required? If you think it is not sufficient, we will of course modify it. If you think it is sufficient, we will simply remove the red colors and directly submit the deanonymized camera ready version of the manuscript
>
> Thank you again for your time.

---

> > ### Comment · Action_Editor_2ZxQ · 2025-04-30
> >
> > **Point 1**: This concern is not addressed to me:
> >
> > *A fundamental weakness of the work is that a version of the front-door identifiability criterion has already been established in a very similar class of time series summary graphs by Eichler and Didelez (2010). They establish both back-door and front-door identification criteria. (The front-door result is their Theorem 4.12. A version of the result was stated without proof in an earlier conference paper, Eichler and Didelez (2007), which is cited by the authors of this manuscript, but this result is not mentioned and the 2010 paper where all details are spelled out is not cited at all.) I say that this result is "similar" but not "exactly the same" because Eichler and Didelez use a slightly different graphical set-up*
> >
> > The 2010 paper is still not cited at all. I would like a paragraph in the introduction discussing the difference from the 2010 paper. This should be very clear, possibly starting with **Difference with Eichler and Didelez (2007, 2010): **. Then I'd like a more technical comparison in Section 3. Right now, the revision does not cite the 2010 result. As this is very close to the contribution, I'd like to see it thoroughly discussed.
> >
> > Point 2 and 3: ok as they are.

---

> > > ### Author Response · Authors · 2025-05-13
> > >
> > > I was initially a bit confused because the paper by Eichler and Didelez was already cited in the second version of the manuscript but then I realized that I referred to it as Eichler and Didelez (2009), based on the issue date, rather than 2010, which is the official publication date. That said, I understand the reviewer's request for a more detailed comparison between our work and theirs. In response, we have updated the citation to Eichler and Didelez (2010) and made an additional round of revisions to the manuscript: our results are now explicitly compared with theirs in the introduction, in Section 3, and again at the end of Section 4.